# Random Projection-Induced Gaussian Latent Features for Arbitrary Style Transfer

**Weizhi Lu[1,3], Zhongzheng Li[1], Dongchen Gao[2], Mingrui Chen[1], Weiyu Li[2,4,*], Jinglin Zhang[1,3,*], Wei Zhang[1,3]**
[1]*School of Control Science and Engineering, Shandong University, China*
[2]*Zhongtai Securities Institute for Financial Studies, Shandong University, China*
[3]*Key Laboratory of Machine Intelligence and System Control, Ministry of Education, China*
[4]*Shandong Blue Ocean Pilot Big Data Development Company Limited, China*

**Reviewed on OpenReview:** https://openreview.net/forum?id=XBu6iqHof8

## Abstract

The feature transfer technique centered on mean and variance statistics, widely known as AdaIN, lies at the core of current style transfer research. This technique relies on the assumption that latent features for style transfer follow Gaussian distributions. In practice, however, this assumption is often hard to meet, as the features typically exhibit sparse distributions due to the significant spatial correlation inherent in natural images. To tackle this issue, we propose first performing a random projection on the sparse features, and then conducting style transfer on these projections. Statistically, the projections will satisfy or approximate Gaussian distributions, thereby better aligning with AdaIN's requirements and enhancing transfer performance. With the stylized projections, we can further reconstruct them back to the original feature space by leveraging compressed sensing theory, thereby obtaining the stylized features. The entire process constitutes a projection-stylization-reconstruction module, which can be seamlessly integrated into AdaIN without necessitating network retraining. Additionally, our proposed module can also be incorporated into another promising style transfer technique based on cumulative distribution functions, dubbed EFDM. This technique faces limitations when there are substantial differences in sparsity levels between content and style features. By projecting both types of features into dense Gaussian distributions, random projection can reduce their sparsity disparity, thereby improving performance. Experiments demonstrate that the aforementioned performance improvements can be achieved on existing state-of-the-art approaches.

## 1 Introduction

Style transfer aims to transfer the style of one image to another while preserving the semantic content information of the latter. Recent research has shown that this objective can be efficiently realized within a deep encoder-decoder framework, by statistically extracting and integrating style and content information from deep convolutional features (Gatys et al., 2016). In this approach, the statistical modeling of style features plays a pivotal role, yet it remains a challenging issue due to the inherent subjectivity and diversity in defining style features.

There are two fundamental statistical approach to model style features: One approach is grounded in the utilization of mean and variance, famously known as AdaIN (Huang & Belongie, 2017), while the other leverages the cumulative distribution function, termed EFDM (Zhang et al., 2022). Despite their impressive performance, both approaches have inherent limitations. This is because deep convolutional features often exhibit sparse distributions (Mahendran & Vedaldi, 2015; Qin et al., 2020), as exemplified in Appendix A.1, and such distributions are unfavorable for effective style transfer. The underlying reasons are as follows. Let us first analyze the AdaIN approach, which operates on the assumption that deep features have Gaussian distributions, rather than sparse distributions. For a style feature map that exhibits a sparse distribution, its style information is typically concentrated within a few large-magnitude feature elements. However, as the number of small-magnitude feature elements increases, the influence of these large

---

*Corresponding Authors: Weiyu Li (liweiyu@sdu.edu.cn) and Jinglin Zhang (jinglin.zhang@sdu.edu.cn).

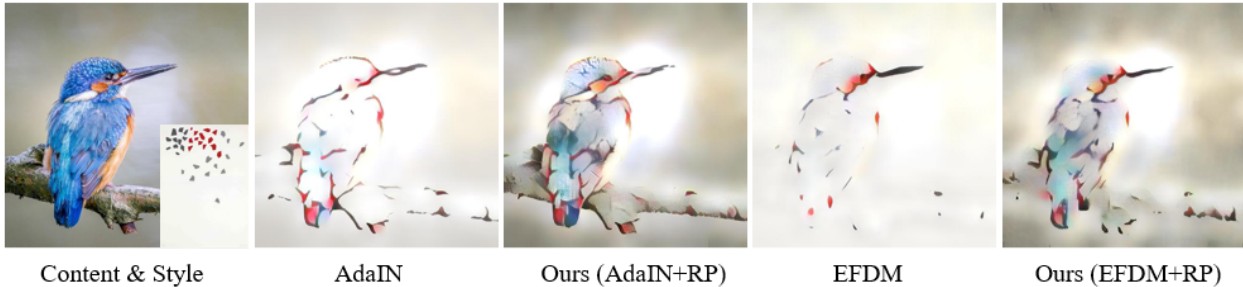

| Content & Style | AdaIN | Ours (AdaIN+RP) | EFDM | Ours (EFDM+RP) |

Figure 1: Given a style image composed of few color spots, both AdaIN (Huang & Belongie, 2017) and EFDM (Zhang et al., 2022) lose key content features, such as the feathers of the bird and the contour of the branch, whereas these features are preserved relatively well when incorporating our random projection (RP) module.

elements on both the mean and variance (essential for AdaIN) tends to decrease. This property suggests that the effect of style transfer with AdaIN may deteriorate, when the style feature map becomes sparser, as demonstrated by the example in Figure 1. In the EFDM method, the cumulative distribution function is adopted to represent style features. The function is transferred by substituting the sorted-feature-elements of the content feature map with corresponding elements from the style feature map. Nevertheless, this method tends to perform worse when the sparsity of the feature distributions is inconsistent between the content feature map and the image feature map. This problem is exemplified in Figure 1, where the style feature map is much sparser than the content feature map. It can be seen that the stylized image experiences noticeable content loss, since the content feature map has to lose a few large-magnitude feature elements, after adopting the elements from the sparser style feature map. Overall, the aforementioned two problems indicate that the sparse distributions of deep features limit the application effects of AdaIN and EFDM.

To address the challenge, in the paper we propose applying random projection to deep features, before performing style transfer on them. Random projection is a technique that projects high dimensional data to low dimensional subspaces by multiplying the data with a random matrix. Statistically, when the entries of random matrices are drawn from Gaussian distributions, the projections will adhere to Gaussian distributions. Similarly, when the matrix entries are sampled from other sparse distributions, such as $\{0, 1\}$ and $\{0, \pm 1\}$ distributions, the projections will approximate Gaussian distributions by the central limit theorem. The projection to lower dimensions can also yield distributions closer to Gaussian. (Meckes, 2012). These properties suggest that random projection enables features to better meet the requirements of AdaIN. Moreover, the dense, Gaussian distribution is favorable to reducing the disparity in sparsity levels between the style image map and the content image map. This, in turn, mitigates the aforementioned sparsity-imbalance issue that can arise in the context of EFDM. By conducting style transfer on the projections of content and style feature maps, we will obtain the stylized projection. To feed the stylized projection into the decoder without network retraining, we further propose to reconstruct the stylized projection back to the original feature map space. This reconstruction relies on compressed sensing theory (Foucart & Rauhut, 2013), which states that a sparse signal can be approximately reconstructed from its random projections, even in the presence of noise. In the context of our research, the stylized projection can be viewed as the projection of content features, perturbed by the projection of style features via style transfer. Then, reconstruction from the stylized projection should yield the desired stylized feature map, which is primarily characterized with content features while being complemented by style features.

The proposed random projection-based style transfer functions as a flexible plug-and-play module, which can be seamlessly integrated into encoder-decoder frameworks for style transfer, without the need for network retraining. Without loss of generality, our research will focus on incorporating this module into two fundamental models: AdaIN (Huang & Belongie, 2017), and EFDM (Zhang et al., 2022). The two models serve as the basis for style transfer and have been employed in most existing style transfer approaches, including attention mechanism-based models like AdaAttN (Liu et al., 2021) and diffusion model-based approaches such as StyleID (Chung et al., 2024). If performance improvements can be realized with these two fundamental models, it is reasonable to anticipate similar improvements in other more advanced models that incorporate them. This is validated in our experiments, where our random projection module indeed empowers existing approaches to overcome the limitations in handling sparse features, thereby achieving notable improvements in preserving content details, enriching and diversifying style elements, and elevating the overall perceptual quality.

## 2    Related Work

**Statistical modeling of style features.**    In the early research of style transfer, Gram matrices are utilized to represent the distribution of style features, achieving impressive results but incurring a relatively high computational burden. To alleviate this issue, AdaIN (Huang & Belongie, 2017) introduces a style modeling approach based on mean and variance, which offers comparable performance but significantly reduces complexity. Consequently, this approach has found wide application in encoder-decoder-based models (Jing et al., 2020; Chandran et al., 2021; An et al., 2020; Lin et al., 2021; An et al., 2021; Liu et al., 2021), as well as in generative adversarial models (Karras et al., 2019; 2020). To further address the limitation of AdaIN in dealing with sparse features, EFDM (Zhang et al., 2022) has recently proposed modeling style features using cumulative distribution functions. This approach reports more pronounced results than AdaIN, attracting increasing attention in recent research (Kwon et al., 2024; Ge et al., 2024; Zhang et al., 2024b). However, as previously noted, EFDM encounters challenges when content and style feature distributions differ in sparsity levels. In the paper, we will demonstrate that our random projection-based feature sampling module can efficiently tackle the two problems present in AdaIN and EFDM.

**Style transfer with global and local features.**    The primary goal of style transfer is to achieve a desired balance between content and style information. Achieving this goal necessitates a delicate manipulation of both global and local feature transfers. Global features are crucial in maintaining content quality and ensuring style consistency, whereas local features are instrumental for refining local details. Initially, research primarily centers on global, channel-wise deep features, as evidenced in works such as (Gatys et al., 2015a;b; 2016; Dumoulin et al., 2016; Johnson et al., 2016; Ulyanov et al., 2016; Gatys et al., 2017; Risser et al., 2017; Huang & Belongie, 2017; Li et al., 2017; 2019; 2020; Huang & Gupta, 2020; Zhang et al., 2022). Subsequently, the focus shifts towards the modeling and transferring on local patches or patterns, as demonstrated in (Chen & Schmidt, 2016; Gu et al., 2018; Zhang et al., 2019; Wang et al., 2021; Hong et al., 2023; Park & Lee, 2019). The transition has driven the emergence of attention-based approaches (Chen et al., 2021; Liu et al., 2021; Wu et al., 2021; Deng et al., 2022; Huang et al., 2023; Xu et al., 2023; Zhu et al., 2023; Ma et al., 2023; Zhang et al., 2024a). Recently, diffusion model-based approaches have significantly boosted style manipulation capabilities. Specifically, methods such as Prompt-to-Prompt (Hertz et al., 2022) and Plug-and-Play (Tumanyan et al., 2023) utilize cross-attention and self-attention maps to guide spatial arrangements during text-driven image editing. More recently, StyleID (Chung et al., 2024) has pushed this paradigm further by integrating key and value features from style images into self-attention layers, facilitating training-free artistic style transfer with precise spatial and statistical alignment. Despite these advancements, efficiently modeling and balancing between global and local features remain challenging, due to the elusive nature of style definition and perception, as well as the complexity of style transfer. In the paper, we achieve a flexible balance between global and local feature transfers, by leveraging the local patch sampling with random projection and the feature grouping on projections.

## 3    Methodology

As discussed before, the proposed random projection-based style transfer can be incorporated into arbitrary encoder-decoder framework, without the need for network retraining. In this section, we elaborate this incorporation process with the fundamental approach AdaIN, as illustrated in Figure 2, and the incorporation into other approaches, like EFDM, StyleID and AdaAttN, can be realized in the similar way, as outlined in Appendices A.2 and A.3.

In Figure 2, we provide the overview of incorporating the random projection module into AdaIN, which mainly consists of three steps. First, the feature maps of the content and style images are projected to low-dimensional spaces by random projection. Then style transfer is conducted over the projections of the two kinds of features. To achieve a balance between local details and global consistency, we here divide each projection into a number of subgroups and conduct style transfer on each subgroup. Finally, with the stylized projection, the target image feature map is derived by sparse reconstruction. Overall, the above three steps constitute a projection-stylization-reconstruction module, which is detailed in subsequent parts.

### 3.1   Random projection

Prior to introducing random projection, let us first review the feature feed-forward process involved in the encoder-decoder model, as illustrated in Figure 2. The model takes as inputs a content image $I_c$ and an arbitrary style image

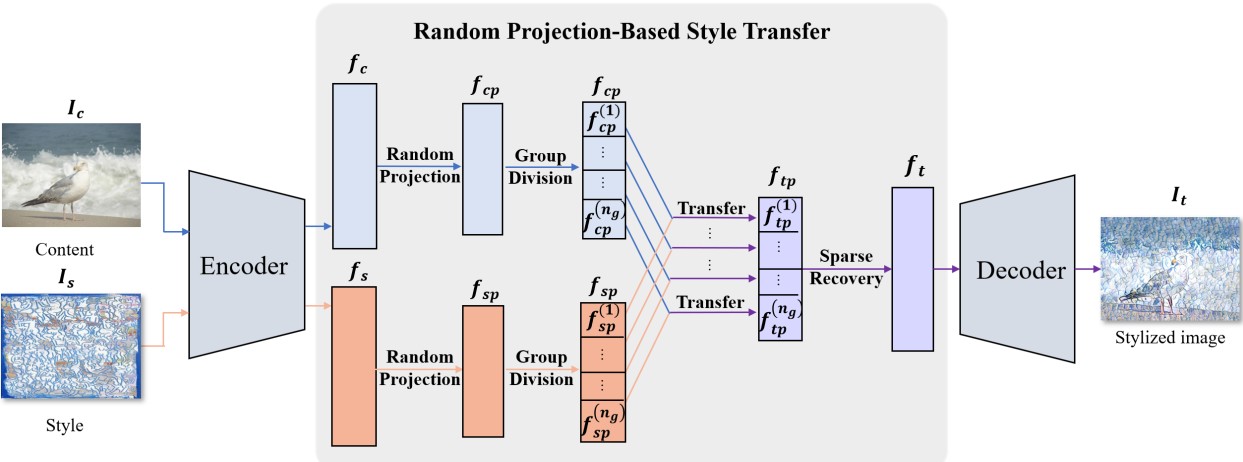

Figure 2: Overview of the random projection-based style transfer implemented on AdaIN. By feeding the content image $I_c$ and the style image $I_s$ into the encoder, we derive the content feature map $f_c$ and the style feature map $f_s$, which are further projected to low dimensional spaces $f_{cp}$ and $f_{sp}$ by random projection. The projections are divided into a number $n_g$ of subgroups: $f_{cp} = [f_{cp}^{(1)}, \cdots, f_{cp}^{(n_g)}]$ and $f_{sp} = [f_{sp}^{(1)}, \cdots, f_{sp}^{(n_g)}]$; and style transfer is conducted on each pair of subgroups $f_{cp}^{(i)}$ and $f_{sp}^{(i)}$, resulting in the stylized projection $f_{tp} = [f_{tp}^{(1)}, \cdots, f_{tp}^{(n_g)}]$. With the projection $f_{tp}$, by sparse reconstruction we derive its counterpart in the original feature space, namely the target feature map $f_t$. Feeding $f_t$ into the decoder, the desired target (stylized) image $I_t$ is finally obtained.

$I_s$, and outputs a target image $I_t$ that combines the semantic content of $I_c$ and the style of $I_s$. This combination is performed in a single or a few layers of the encoder. In each layer, we can derive two feature maps respectively for $I_c$ and $I_s$. Usually, each feature map will consist of multiple channels. For $I_c$, we denote the feature map in each channel with a vector $f_c \in \mathbb{R}^{(H_c \times W_c) \times 1}$, where $H_c$ and $W_c$ indicate the height and width of the content feature map. In the similar way, we can define $f_s \in \mathbb{R}^{(H_s \times W_s) \times 1}$ for $I_s$. Note that we here describe the feature of each channel, rather than the features of all channels, since the random projection-based style transfer is conducted on channel-wise basis. Given the content feature map $f_c$ and the style feature map $f_s$, we then can derive their random projections

$$f_{cp} = A_c f_c \text{ and } f_{sp} = A_s f_s \tag{1}$$

where $A_c \in \mathbb{R}^{M_c \times (H_c \times W_c)}$ and $A_s \in \mathbb{R}^{M_s \times (H_s \times W_s)}$ are the random projection matrices.

For random projection, the choice of random matrices is crucial. In this paper, we investigate four commonly-used random matrices in compressed sensing: the {0,1}-matrix with $\sqrt{M_c}$ nonzero elements per column (Lu et al., 2018), the {1,-1}- and {0,1,-1}-matrices with elements taken with equal probabilities (Achlioptas, 2003; Amini & Marvasti, 2011), and the Gaussian matrix (Candes & Tao, 2005). It can be seen that the {0,1}-matrix exhibits very sparse structures compared to other three kinds of matrices. Empirically, the three relatively dense matrices perform well when the style feature map exhibits very sparse distributions. This is because the random projection based on dense matrices is more likely to approximate Gaussian distributions. In contrast, when the style feature is not very sparse, the sparse {0,1}-matrix tends to perform better. The advantage may be explained with the following fact. Essentially, the {0,1}-matrix based random projection performs a random, sparse sampling over the features $f_c$ and $f_s$. The resulting projections $f_{cp} \in \mathbb{R}^{M_c}$ and $f_{sp} \in \mathbb{R}^{M_s}$ have their each element related to a few elements of the original features $f_c$ and $f_s$, reflecting the local correlation of original features. This allows the stylization on feature projections to better capture the local details compared to stylization on original features.

In addition to the distribution of random matrices, style transfer is also related to the compression rate of random matrices, which can be written as $r = M_c/(H_c \times W_c)$. With the decreasing of compression rate $r$, the following sparse reconstruction will become hard and then introduce noise. Empirically, the increased noise tends to degrade the quality of content, but enrich the diversity of style. Moreover, the random variation of the random matrices themselves can also result in diverse styles.

### 3.2 Grouping-based style transfer

With the feature projections $f_{cp} \in \mathbb{R}^{M_c}$ and $f_{sp} \in \mathbb{R}^{M_s}$ derived in the previous random projection, we are ready to perform style transfer on them. To control the impact of style transfer on local details, we propose to first divide the feature projections into a number $n_g$ of subgroups, namely having $f_{cp} = [f_{cp}^{(1)}, \cdots, f_{cp}^{(n_g)}]$ and $f_{sp} = [f_{sp}^{(1)}, \cdots, f_{sp}^{(n_g)}]$, and then perform style transfer on each pair of subgroups $f_{cp}^{(i)} \in \mathbb{R}^{\frac{M_c}{n_g}}$ and $f_{sp}^{(i)} \in \mathbb{R}^{\frac{M_s}{n_g}}$. It is evident that the greater the number $n_g$ of subgroups, the fewer local elements are involved in style transfer. By adjusting the value of $n_g$, we can strike a balance between local details and global consistency.

By conducting AdaIN on each pair of subgroups $f_{cp}^{(i)}$ and $f_{sp}^{(i)}$, the stylized feature is derived as

$$f_{tp}^{(i)} = \sigma(f_{sp}^{(i)}) \frac{f_{cp}^{(i)} - \mu(f_{cp}^{(i)})}{\sigma(f_{cp}^{(i)})} + \mu(f_{sp}^{(i)}). \tag{2}$$

Combining the results $f_{tp}^{(i)} \in \mathbb{R}^{\frac{M_c}{n_g}}$ of all subgroups, we can obtain the stylized feature on the entire projection $f_{tp} = [f_{tp}^{(1)}, \cdots, f_{tp}^{(n_g)}] \in \mathbb{R}^{M_c}$.

Similarly as the content projection $f_{cp}$ and the style projection $f_{sp}$ as derived in (1), we here hypothesize that the stylized projection $f_{tp}$ is a random projection of an underlying target (stylized) feature map $f_t$, namely

$$f_{tp} = A_c f_t. \tag{3}$$

Here, we exploit the same projection matrix $A_c$ with the content projection $f_{cp}$. This choice is based on the fact that the stylized projection $f_{tp}$ is predominantly characterized with the content projection $f_{cp}$, and is slightly altered by the style projection $f_{sp}$ via style transfer. According to compressed sensing theory (Foucart & Rauhut, 2013), a target feature map $f_t$ similar to the content feature map $f_c$ can be reconstructed through sparse reconstruction, if the noise introduced to the content projection $f_{cp}$, namely the variation induced by style transfer, is limited. In general, the variation in $f_{cp}$ during style transfer is more significant when using EFDM compared to AdaIN. The difference arises because AdaIN changes only the mean and variance of $f_{cp}$, whereas EFDM modifies the cumulative distribution function by replacing all elements of $f_{cp}$ with those of $f_{tp}$.

### 3.3 Sparse Reconstruction

With the projection hypothesis (3), we now need to reconstruct the target image feature map $f_t$ from its projection $f_{tp}$. By compressed sensing, $f_t$ can be approximately derived by

$$\hat{f}_t = \arg\min_{f_t} \|A_c f_t - f_{tp}\|_2^2 + \lambda \|f_t\|_1, \tag{4}$$

if the feature $f_t$ has adequately sparse distributions, and the projection matrix $A_c$ has sufficiently low correlations between columns. The two conditions should be approximately met, as the deep convolutional feature $f_t$ is usually sparse, and the random matrices employed here are commonly utilized in compressed sensing. After obtaining the target feature map $f_t$, we can further feed it into the encoder-decoder model to generate the desired target image $I_t$.

As a convex problem, (4) can be tackled with standard optimization algorithms. Given that deep convolution features usually have high dimensions, selecting an efficient algorithm is essential. Commonly used algorithms include Fast Iterative Shrinkage-Thresholding Algorithm (FISTA) (Beck & Teboulle, 2009) and Orthogonal Matching Pursuit (OMP) (Lubonja et al., 2024). In our experiments, we adopt FISTA due to its lower complexity while maintaining performance comparable to that of OMP, as demonstrated in Appendix A.4. FISTA supports GPU-accelerated parallel computing (Feinman, 2021), thus ensuring high computational efficiency. For instance, it typically takes less than two seconds to reconstruct a single feature map, when executed on a GeForce RTX 4090 GPU. The time consumption issue is comprehensively analyzed in Appendix A.5.

## 4 Experiments

Based on the analysis presented in the previous section, our random projection-based style transfer method is subject to the influence of several parameters tied to random projection. These include the distribution and compression rate

$r$ of random matrices, as well as the number $n_g$ of subgroups into which the feature projections $f_{cp}$ and $f_{sp}$ are divided. In this section, we will first examine how these parameters impact stylized results, enabling us to achieve a desired balance between style and content by reasonably adjusting the parameters. Then, we will showcase the performance improvements achieved by integrating our approach into two fundamental style transfer models: AdaIN (Huang & Belongie, 2017) and EFDM (Zhang et al., 2022), as well as into their advanced variants and other state-of-the-art approaches, such as the diffusion model-based StyleID (Chung et al., 2024) and the attention mechanism-based AdaAttN (Liu et al., 2021). Prior to these studies, we will first outline the implementation details.

## 4.1 Implementation Details

Given the models AdaIN, AdaAttN, EFDM and StyleID, as illustrated in Figure 2, our random projection-based style transfer module is incorporated into each layer of these models that involve style transfer, without the need for network retraining. The computation cost introduced by our module primarily stems from the sparse reconstruction outlined in (4), which is implemented using the FISTA algorithm (Beck & Teboulle, 2009; Feinman, 2021). For the algorithm, we assign the regularization parameter of $\lambda = 0.5$ and limit the maximum number of iterations to 10. Regarding random projection, we generally set the compression rate $r = M/N = 0.8$ or 1. For style transfer, we set the number of subgroups within feature projections to $n_g = 1$. Empirically, our performance is insensitive to minor variations in these parameters, and no particularly meticulous parameter tuning is required. The selection and impact of these parameters are discussed in Appendices A.6 and A.7. The code is publicly available[1].

## 4.2 Impact of the parameters of random projection on style transfer

**Distribution of random matrices.** In Figure 3, we compare the stylized results derived with four popular random projection matrices: {0, 1}, {-1, 0, 1}, {-1, 1}, and Gaussian matrices. It is evident that when the style feature exhibits a sparse distribution, as illustrated in the first row, the {0, 1}-matrix notably underperforms compared to the other three matrices, while the latter three exhibit comparable performance. Notably, the {0, 1}-matrix comprises only a ratio of $1/\sqrt{M_c}$ nonzero entries, leading to much sparser distributions than the other matrices. Consequently, it struggles in projecting the style feature towards denser, Gaussian distributions, resulting in inferior style transfer performance. Conversely, the {0, 1}-matrix excels when the style feature displays relatively dense distributions, as depicted in the second row of Figure 3. In such scenarios, generating projections towards Gaussian distributions becomes more feasible. In our experiments, unless otherwise specified, we will employ Gaussian matrices.

**Compression rate of random matrices and the number of subgroups within feature projections.** In Figure 4, we compare the stylized results by varying the compression rate $r$ of random {0, 1}-matrices from 0.2 to 1, and the number $n_g$ of subgroups within feature projections from 1 to 256. It is apparent that as the compression rate $r$ decreases or the subgroup number $n_g$ increases, the style gradually becomes prominent while the content becomes less evident. The impact of compression rates can be understood as follows. In compressed sensing, a lower compression rate $r$ will lead to a higher reconstruction noise in the reconstructed stylized feature $f_t$. Then the increased noise introduces more irregular style patterns, while degrading the content. Regarding the increase in the number of subgroups $n_g$, as mentioned in the previous section, it implies that the number of feature elements within each subgroup for style transfer decreases. This element-restricted style transfer may overly focus on local details and neglect global constraints, leading to content loss. The aforementioned performance trends are also observed in supplementary experiments, as presented in Figures 13-15 of Appendix A.7. Based on these trends, satisfactory performance can generally be achieved by simply setting $r = 1$ and $n_g = 1$.

## 4.3 Improvement of our random projection module over state-of-the-art approaches

**Improvement over two fundamental models: AdaIN and EFDM.** In Figure 5, we examine the style transfer performance by incorporating our random projection (RP) module into AdaIN and EFDM. For the sake of generality, the content images cover four distinct types of objects: human faces, fish sketches, flowers, and buildings; and, the style images are characterized by sparse lines or patches. It is seen that the original models AdaIN and EFDM exhibit inferior performance in terms of content detail preservation and style feature transfer. As discussed previously, this limitation stems from the fact that these two models are not well suited to processing style or content features that

---

[1]https://github.com/lxxnjd/RP-Gaussian-Latent-Features-Style-Transfer-pytorch

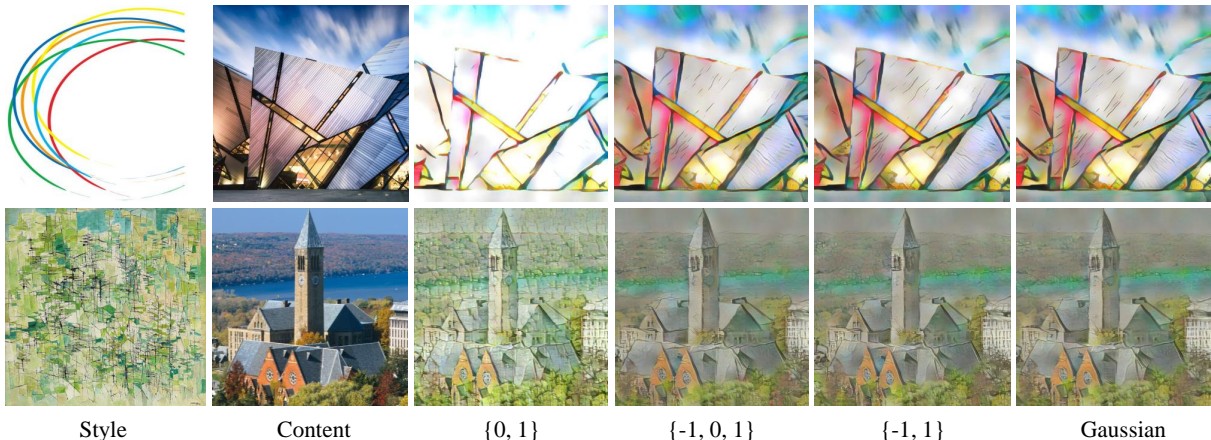

Figure 3: Stylized results derived with four different random matrices: {0, 1}, {-1, 0, 1}, {-1, 1} and Gaussian matrices.

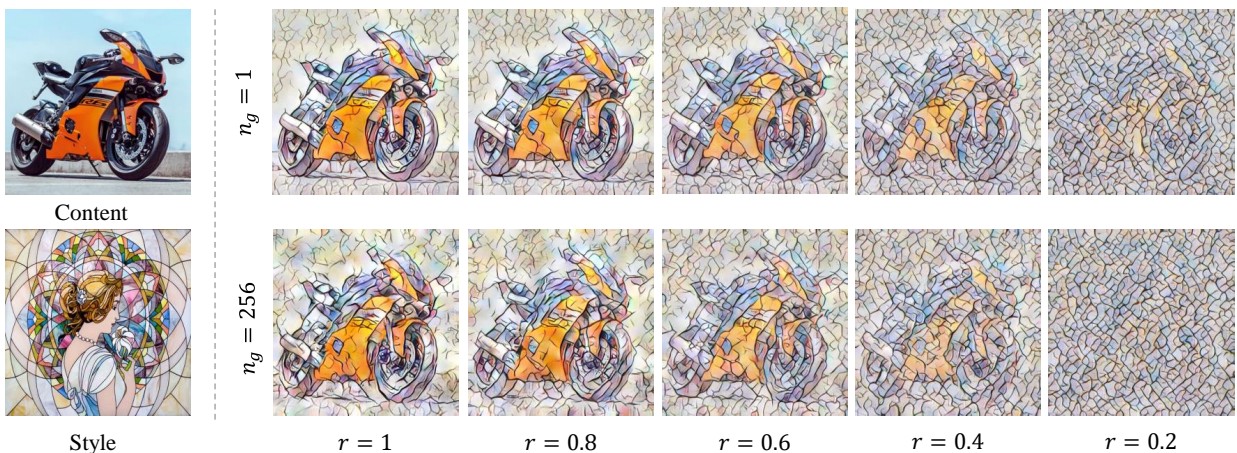

Figure 4: Stylized results derived by varying the compression rate $r$ of random matrices from 1 to 0.2, and the number of subgroups $n_g$ within feature projections from 1 to 256. Random projection is implemented with {0, 1} matrices.

follow sparse distributions. In contrast, by incorporating our random projection module into these two models, the resulting AdaIN+RP and EFDM+RP achieve notable performance improvements by transforming the features from sparse distributions toward Gaussian distributions. Our success with the two foundational models not only validates the effectiveness of our method but also suggests its broad applicability, given that the two models form the core of most existing style transfer approaches, as elaborated later.

**Improvement over the diffusion model-based approach: StyleID.** To validate the effectiveness and generalizability of our method, we examine how it improves the performance of the state-of-the-art approach StyleID by keeping either the content or the style constant while varying the other, as depicted in Figures 6 and 7. In Figure 6, a facial image undergoes stylization with various style images. It is evident that across the various styles, our method consistently brings about significant enhancements in texture details and contrast, color saturation, and overall visual perception. Similar improvements are also noticeable in Figure 7, where various content images are stylized using a black-lined portrait sketch featuring red lips. Particularly, it can be seen that our method successfully and significantly transfers the color of the red lips, even though this color constitutes only a very small portion of the entire style image. This validates the advantage of our random projection module in capturing minor but crucial sparse features.

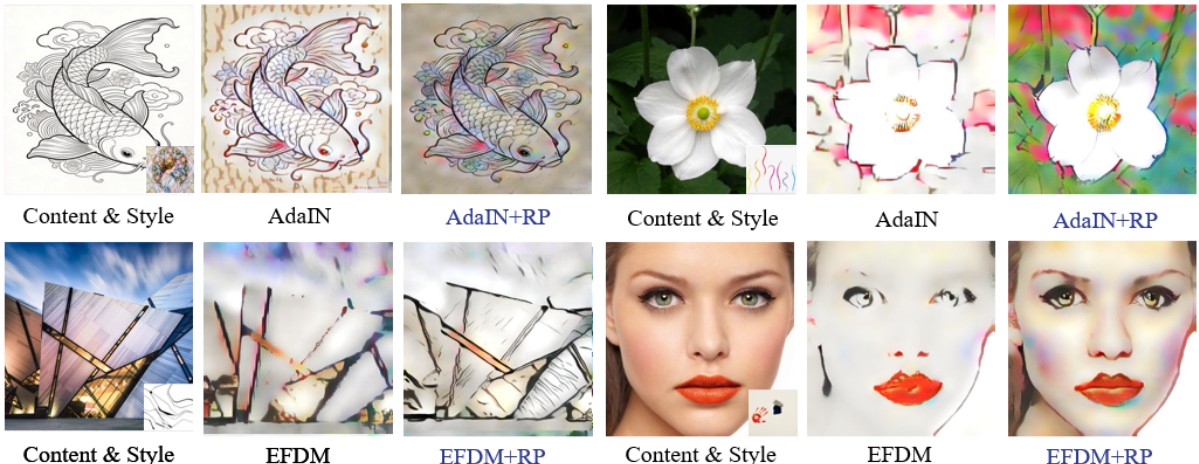

Figure 5: Stylized results derived by incorporating vs. *not* incorporating our random projection (RP) module into the two fundamental models: AdaIN and EFDM.

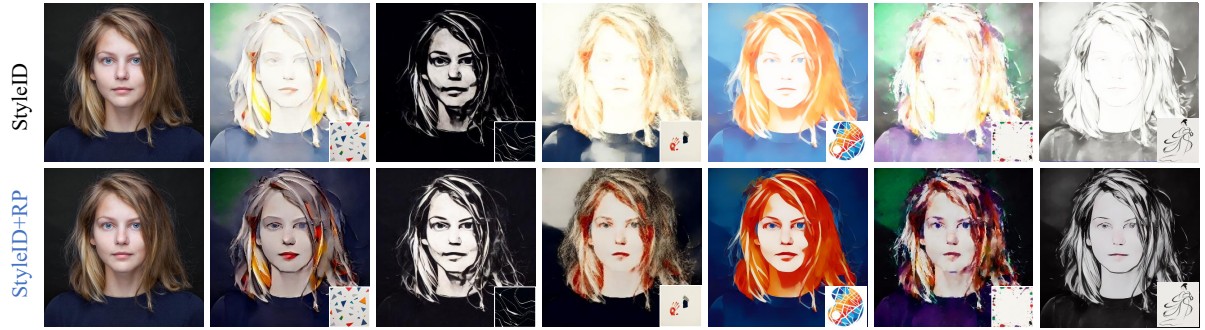

Figure 6: Stylized results derived by incorporating vs. *not* incorporating our random projection (RP) module into the diffusion model-based approach: StyleID, with a fixed content but various styles.

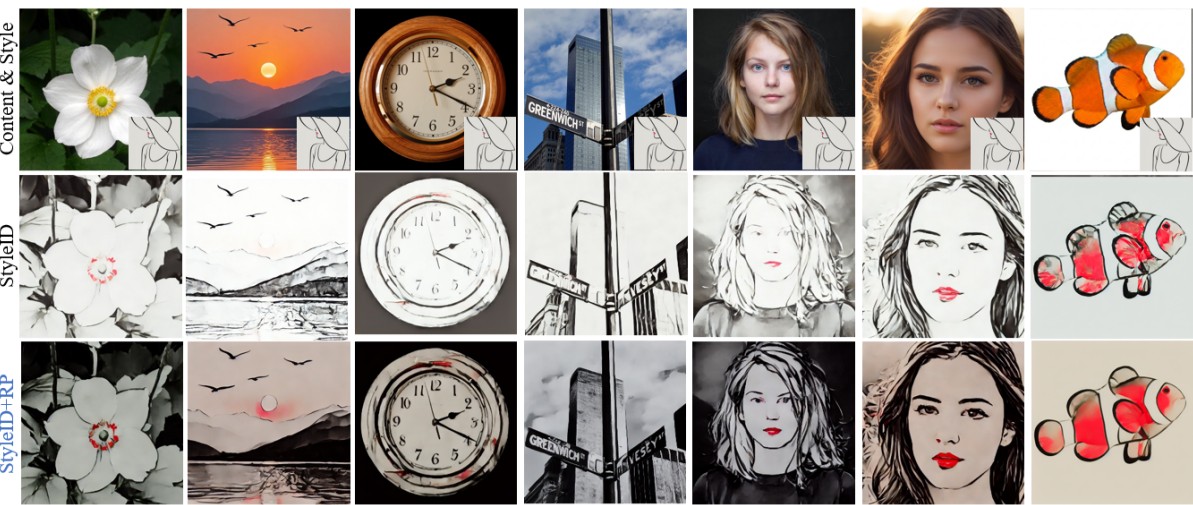

Figure 7: Stylized results derived by incorporating vs. *not* incorporating our random projection (RP) module into the diffusion model-based approach: StyleID, with various contents but a fixed style (a black-line portrait with red lips).

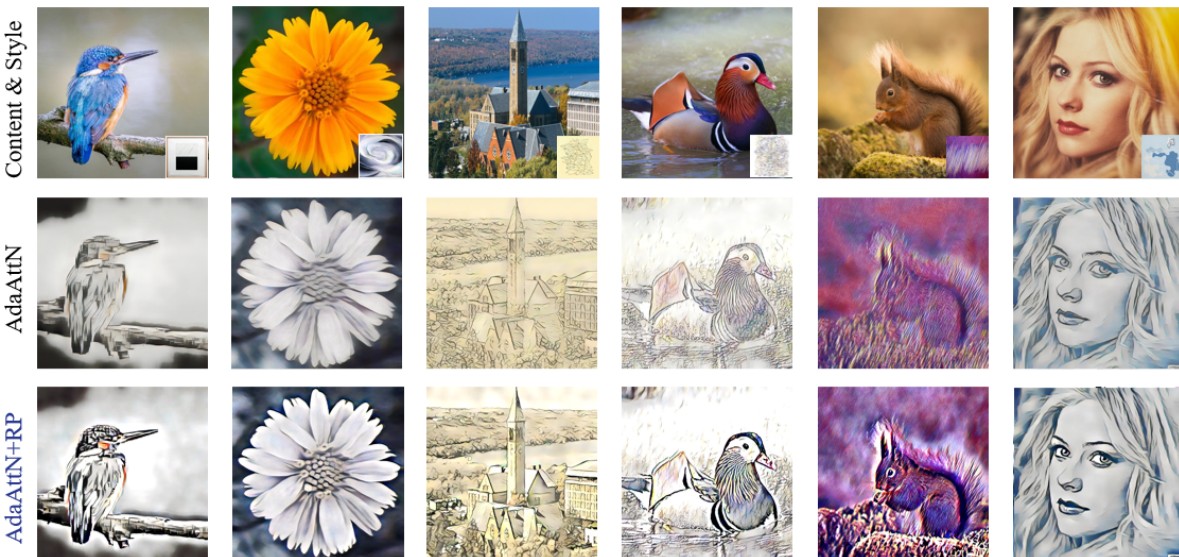

Figure 8: Stylized results derived by incorporating vs. *not* incorporating our random projection (RP) module into the attention-based approach: AdaAttN.

Table 1: Quantitative results of LPIPS, ArtFID and User Study. The better results are marked with an underline.

| Metric | AdaIN | AdaIN+RP | EFDM | EFDM+RP | StyleID | StyleID+RP |
|---|---|---|---|---|---|---|
| LPIPS $\downarrow$ | 0.665 | 0.579 | 0.686 | 0.592 | 0.650 | 0.568 |
| ArtFID $\downarrow$ | 31.01 | 29.65 | 33.88 | 31.37 | 28.51 | 26.42 |
| Preference ratio$\uparrow$ | 0.26 | 0.74 | 0.19 | 0.81 | 0.13 | 0.87 |

**Improvement over the attention-based approach: AdaAttN.** Figure 8 illustrates notable performance improvements in texture clarity and contrast, when integrating our random projection module into the AdaAttN. Furthermore, the resulting AdaAttN+RP also outperforms other well-recognized approaches, including AesPA(Hong et al., 2023), RAST(Ma et al., 2023), StyTr$^2$(Deng et al., 2022), IEST(Chen et al., 2021), Styleformer(Wu et al., 2021), and Artflow(An et al., 2021), as demonstrated in Figures 16-18 of in the Appendix A.8. Note that our AdaAttN+RP performs well not only on visually sparse images but also on visually dense ones. This broader applicability to visually dense images arises from the fact that such images typically exhibit sparse feature distributions due to the spatial continuity and correlation of image pixels. As a result, our random projection module can be used to further enhance the style transfer performance by transforming these sparse feature distributions towards the desired Gaussian distributions.

**Quantitative results.** The results shown in Figures 5-8 demonstrate that our random projection module significantly improves the quality of content, delivering superior perceptual performance in style transfer. This property is further supported by the quantitative results presented in Table 1, where LPIPS (Zhang et al., 2018) measures the content fidelity of stylized results, and ArtFID (Wright & Ommer, 2022) assesses the overall style transfer performance by considering both content and style qualities. Moreover, we conducted a user study with 50 participants who selected their preferred stylized results from 60 trials based on overall perceptual quality. All these results validate the advantage of our approach.

**Potential limitation.** The main advantage of our random projection module is that it can prevent the loss of critical features in either content or style images, when their latent features exhibit extremely sparse distributions. Typical examples include the body contour (content features) of the bird in Figure 1 and the red lips (style features) of the lady in Figure 7. In practical scenarios, however, the feature loss of content or style caused by existing style transfer approaches may be imperceptible to human vision, resulting in an unobvious performance gain of our method. Nevertheless, even without improved perceptual quality, our module will not degrade the quality, since compressed sensing

enables reliable feature reconstruction from random projections when the compression rate is not extremely low. This indicates that our module can be universally integrated into existing style transfer approaches without concerns about performance degradation.

## 5 Conclusion

Existing style transfer approaches based on the encoder-decoder architecture typically employ either the mean-and-variance strategy (proposed in AdaIN) or the cumulative distribution function (introduced in EFDM) to model the distribution of style features. The two methods implicitly favor the scenarios where the features of both style and content images roughly follow Gaussian distributions. Nevertheless, this requirement is often hard to satisfy, as deep encoder features usually exhibit sparse distributions. Consequently, when handling highly sparse features, the two methods often result in poor perceptual quality, with a significant loss of texture details, contrast, color, and content. In this paper, we have for the first time identified and tackled this issue by incorporating a random projection module into the encoder-decoder architecture, in order to transform features from sparse distributions towards the desired Gaussian distributions. This module operates as a flexible plug-and-play component, without the need for network retraining. When integrated into the fundamental models AdaIN and EFDM, as well as other cutting-edge approaches like the diffusion model-based StyleID and the attention mechanism-based AdaAttN, this module brings about notable performance improvements.

## Acknowledgements

We sincerely thank the editors and anonymous reviewers for their precious time and invaluable suggestions. Furthermore, we gratefully acknowledge the support provided by the National Key Research and Development Program of China (Grant Nos. 2022YFB3206900 and 2023YFA1008701) and the National Natural Science Foundation of China (Grant Nos. 61991412, 61801264, and 12001318).

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

# A  Appendix

## A.1  Sparsity of latent features

In this part, we investigate the sparsity of latent features generated by the encoders of the studied style transfer approaches. These features indeed tend to follow sparse distributions for most images, for two main reasons. First, most images contain relatively large smooth regions, which facilitate the generation of sparse feature distributions. Second, notably, these encoders do not reduce the input image dimensions to a considerably lower level. Specifically, unlike conventional encoders designed for dimensionality reduction, the encoders in AdaIN, AdaAttN and EFDM *increase* the feature dimension from the channel size of [3, 512, 512] to [512, 64, 64] (with vector dimension greater than 2,000,000), whereas the encoder of StyleID reduces the feature dimension from the channel size of [3, 512, 512] to [4, 64, 64] (with vector dimension greater than 10,000). These feature dimensions are substantially larger than those of encoder features from conventional VAEs and GANs, which are usually on the order of hundreds. Such high dimensionality inevitably leads to feature redundancy, thereby resulting in sparse structures.

To quantify the sparsity of these encoder features, as illustrated in Figure 9, we further investigate their *kurtosis*, a canonical statistical metric for quantifying the sparsity of random variables. Kurtosis is defined as $\mathbb{E}(\frac{x_i-\mu}{\sigma})^4$, where $x_i$ denotes the $i$-th elements of a vector $x$, and $\mu$ and $\sigma$ are their mean and standard deviation. A larger kurtosis value indicates a sparser distribution. If the kurtosis value of a variable is larger than that (equal to 3) of Gaussian variables, the variable can be roughly regarded as following a sparse (or called heavy-tailed) distribution. From Figure 9, it is seen that the kurtosis of AdaIN features is higher than that of StyleID features, which can be attributed to their higher feature dimensions. For most images, the kurtosis values of both feature types are notably higher than the Gaussian baseline (i.e., 3), indicating sparse feature distributions.

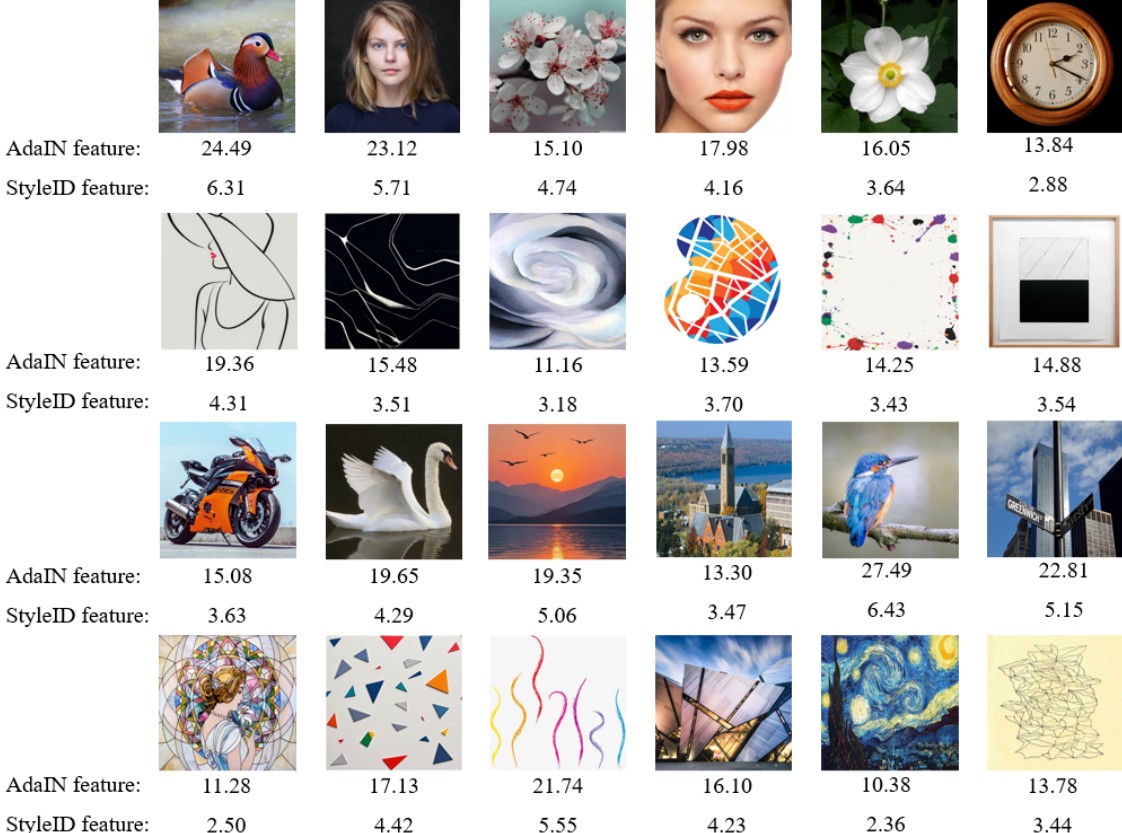

Figure 9: The kurtosis values of the AdaIN and StyleID features extracted from the content and style images studied in the paper. For most images, these values are markedly higher than the Gaussian baseline (i.e., 3), indicating sparse feature distributions.

## A.2 Overview of incorporating random projection into AdaAttN

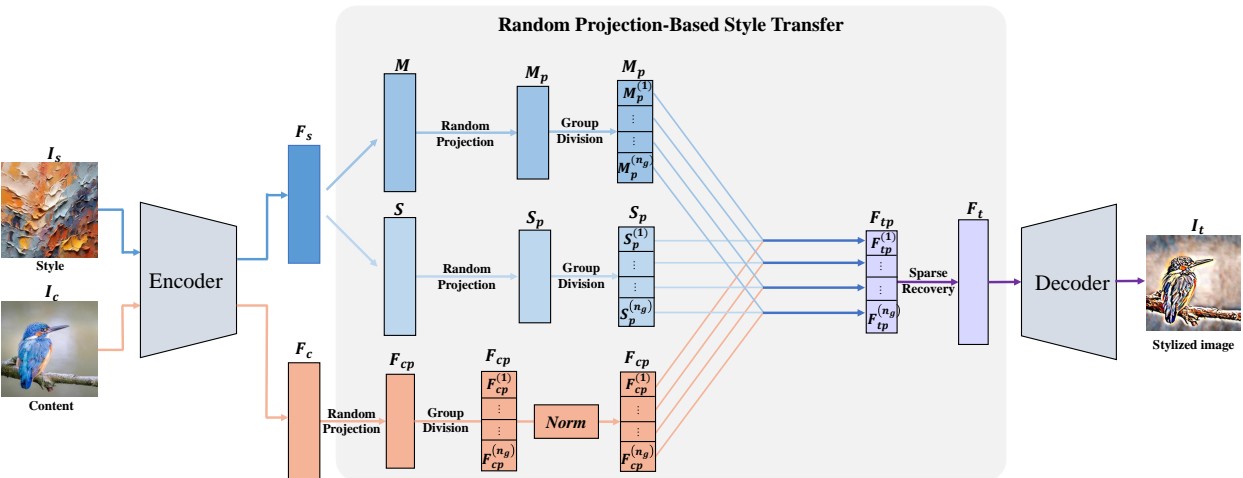

Figure 10: Overview of the random projection-based style transfer implemented on AdaAttN. By feeding the style image $I_s$ and the content image $I_c$ into the VGG encoder, we derive the style feature map $F_s$ and the content feature map $F_c$. Then the weighted mean $M$ and standard variance $S$ are obtained from $F_s$, and the specific method is mentioned in AdaAttN (Liu et al., 2021). $M$, $S$ and $F_c$ are further projected to low dimensional spaces $M_p$, $S_p$ and $F_{cp}$ by random projection. The projections are divided into a number $n_g$ of subgroups: $M_p = [M_p^{(1)}, \cdots, M_p^{(n_g)}]$, $S_p = [S_p^{(1)}, \cdots, S_p^{(n_g)}]$ and $F_{cp} = [F_{cp}^{(1)}, \cdots, F_{cp}^{(n_g)}]$, and $F_{cp}$ is further mean-variance channel-wise normalized, represented by $Norm$ in the figure. Then style transfer is conducted on each pair of subgroups $M_p^{(i)}$, $S_p^{(i)}$ and $F_{cp}^{(i)}$, resulting in the stylized projection $F_{tp} = [F_{tp}^{(1)}, \cdots, F_{tp}^{(n_g)}]$. The formula for style transfer is formulated as $F_{tp}^{(i)} = F_{cp}^{(i)} * S_p^{(i)} + M_p^{(i)}$. With the projection $F_{tp}$, by sparse reconstruction we derive its counterpart in the original feature space, namely the target feature map $F_t$. Feeding $F_t$ into the decoder, the desired target (stylized) image $I_t$ is finally obtained.

## A.3 Overview of incorporating random projection into StyleID

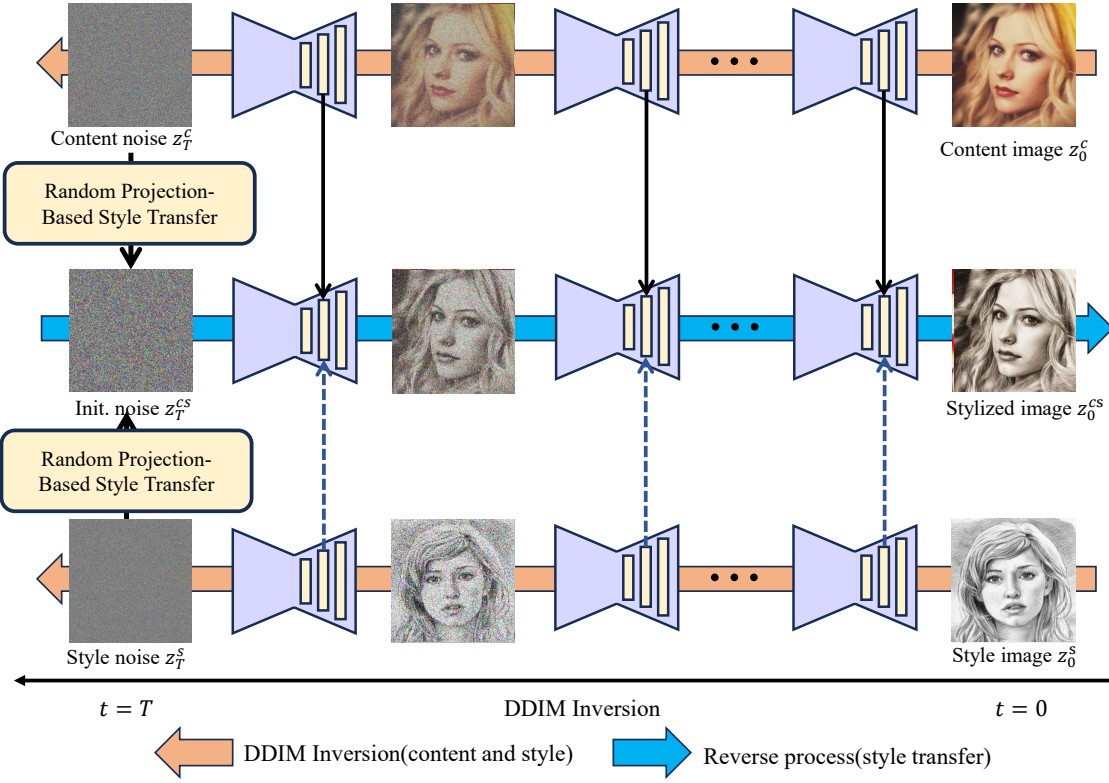

Figure 11: Overview of the random projection-based style transfer implemented on StyleID. First, content and style images undergo identical DDIM inversion process, with attention-based style injection module strictly following StyleID's original implementation. Subsequently, our random projection-based style transfer module operates within StyleID's Initial Latent AdaIN module, performing the operations of random projection and style transfer as illustrated in Figure 2. Finally, the modified latent progresses through StyleID's reverse diffusion process for image synthesis.

## A.4 Comparison of two representative sparse reconstruction algorithms: FISTA and OMP

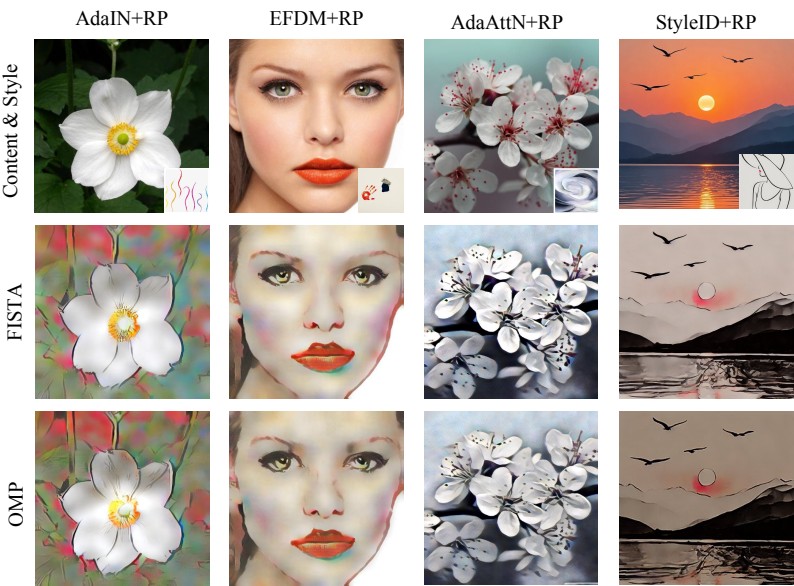

Figure 12: Stylized results derived with two commonly-used sparse reconstruction algorithms: the optimization-based FISTA (Feinman, 2021) vs. the greedy search-based OMP (Lubonja et al., 2024). It is seen that the visual effects of the two algorithms are highly similar.

**A.5  Time consumption of our random projection module**

In Tables 2, 3, and 4, we present the time consumptions of feature encoding, style transfer, and decoding for AdaIN (which shares the same framework as EFDM), AdaAttN, and StyleID implemented on a GeForce RTX 4090 GPU. Additionally, we provide the time costs of random projection and sparse reconstruction involved in our random projection (RP) module. It is evident that the time overhead introduced by our RP module is minimal, particularly when compared to the substantial time savings achieved by eliminating the need for network retraining.

Table 2: Time consumption of AdaIN/EFDM incorporating our RP module

| AdaIN/EFDM | | | Our RP module | |
|---|---|---|---|---|
| Encoding | Style transfer | Decoding | Random projection | Sparse reconstruction |
| 136ms | 18ms | 22ms | 196ms | 203ms |

Table 3: Time consumption of AdaAttN incorporating our RP module

| AdaAttN | | | Our RP module | |
|---|---|---|---|---|
| Encoding | Style transfer | Decoding | Random projection | Sparse reconstruction |
| 872ms | 172ms | 371ms | 2401ms | 2303ms |

Table 4: Time consumption of StyleID incorporating our RP module

| StyleID | | | Our RP module | |
|---|---|---|---|---|
| Encoding | Style transfer | Decoding | Random projection | Sparse reconstruction |
| 10533ms | 10ms | 5603ms | 590ms | 626ms |

## A.6 Selection of key parameters

For ease of parameter tuning and practical application, the selection of key parameters is outlined as follows:

1) **Random projection matrices:** In most cases, Gaussian matrices with zero mean and unit variance deliver satisfactory performance. If Gaussian matrices fail to meet requirements (e.g., for dense content features as illustrated in Figure 3), binary $\{0, 1\}$-matrices may yield better performance.

2) **Compression rate** $r = M/N$**:** According to compressed sensing theory, perfect sparse recovery requires the compression rate to satisfy $M > 2k$, where $k$ denotes the feature sparsity, namely the number of nonzero entries in the $N$-dimensional feature vectors. For style transfer tasks, besides recovery accuracy, we also need to consider the perceptual quality of the stylized results. Empirically, as illustrated in Figure 4 and Figures 13-15, setting $r \geq 0.6$ can usually yield satisfactory performance. A lower compression rate $r$ tends to introduce more recovery errors (i.e. noise), thereby degrading the content feature quality.

3) **Number $n_g$ of subgroups in feature projections:** For simplicity, $n_g$ can be set to 1 in most cases. As observed in Figure 4, increasing $n_g$ from 1 to 256 leads to slight variations in stylized effects, and such variations are far less pronounced than those caused by the compression rate $r$.

4) **Sparse recovery algorithms:** We recommend two commonly-used algorithms: FISTA and OMP. For FISTA, in our experiments the regulation parameter is set to $\lambda = 0.5$ and the iteration number is limited to 10; and for OMP, its sparsity parameter $k$ can be determined based on feature sparsity.

### A.7 Impact of the varying of compression rate $r$ and number $n_g$ of subgroups

In Figures 13 to 15, following the investigation in Figure 4, we further evaluate the performance gains of our random projection module when integrated into AdaAttN, EFDM and StyleID, by decreasing the compression rate $r$ from 1 to 0.2 and increasing the number $n_g$ of subgroups from 1 to 256. The results demonstrate that as the compression rate $r$ decreases, the content quality tends to degrade while the stylized effect tends to be enhanced, owing to the introduction of more recovery errors (i.e. noise). Similarly, increasing the subgroup number $n_g$ slightly boosts stylization effects. Regarding parameter tuning, simply setting $r = 1$ and $n_g = 1$ suffices for our method to achieve notable performance gains over the aforementioned style transfer approaches.

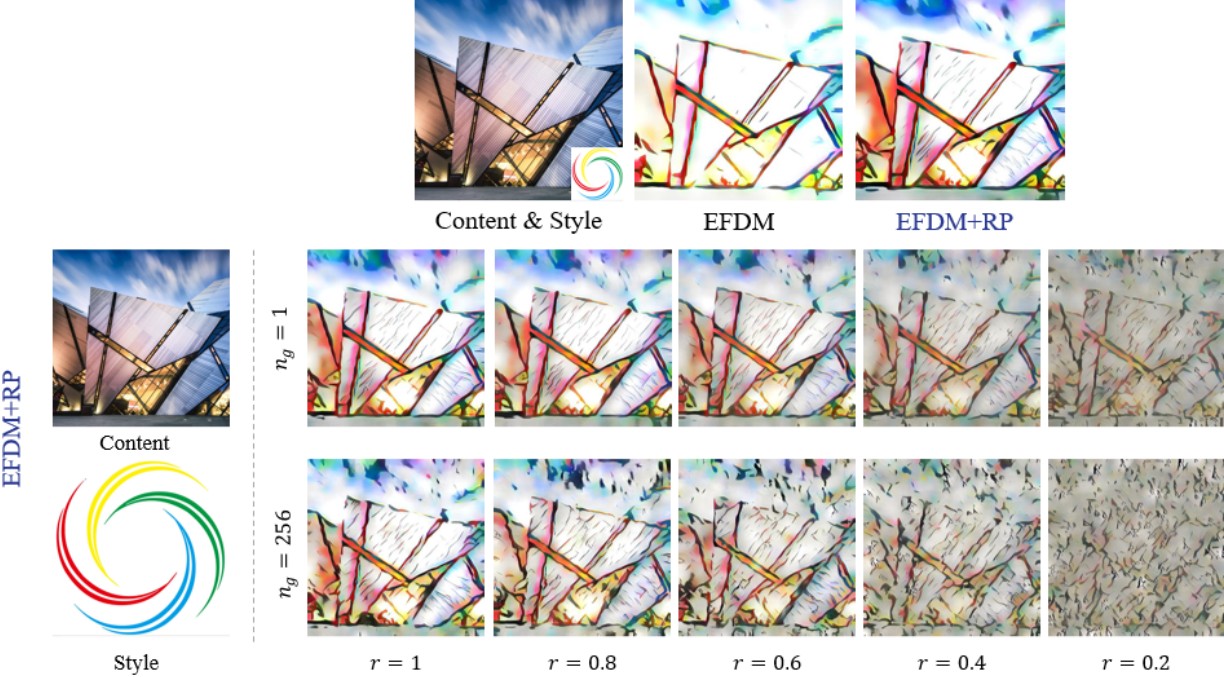

Figure 13: The stylized results derived by integrating our random projection module into EFDM, where the compression rate $r$ is decreased from 1 to 0.2 and the number $n_g$ of subgroups is increased from 1 to 256. Compared to EFDM, our method EFDM+RP exhibits better texture-detail performance, when the compression rate $r$ is sufficiently large.

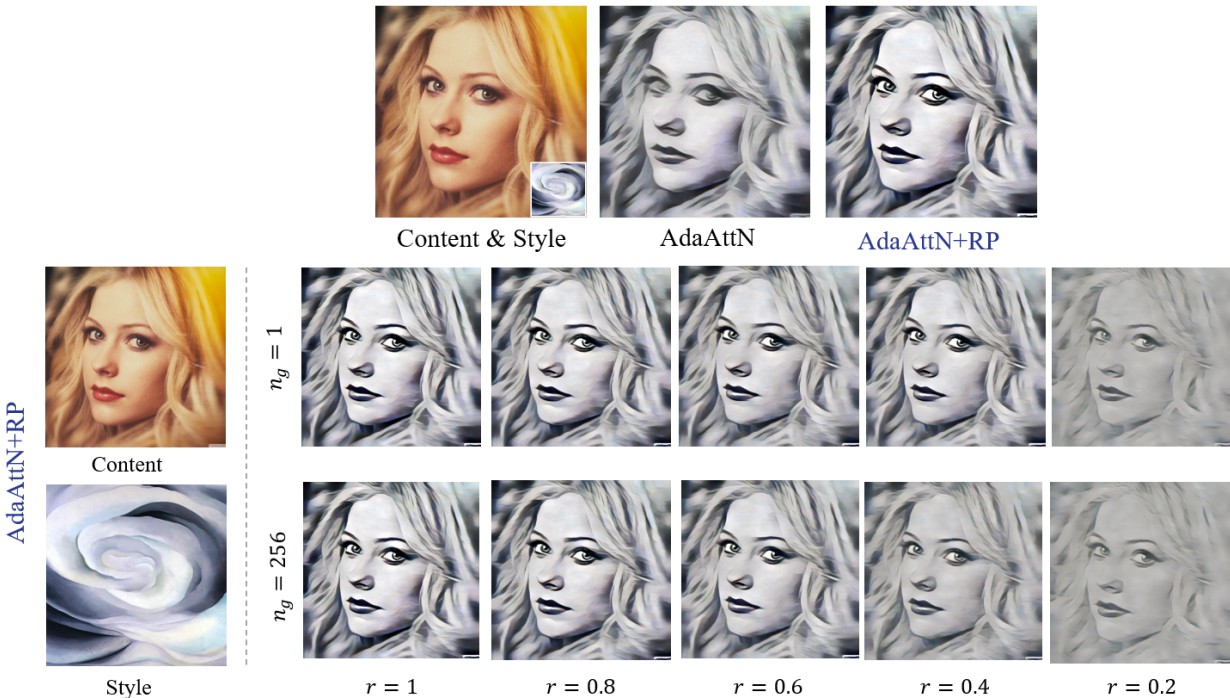

Figure 14: The stylized results derived by integrating our random projection module into AdaAttN, where the compression rate $r$ is decreased from 1 to 0.2 and the number $n_g$ of subgroups is increased from 1 to 256. Compared to AdaAttN, our method AdaAttN+RP achieve markedly superior performance when the compression rate $r$ is sufficiently large, as clearly manifested in the refined details of eyes and individual hair strands.

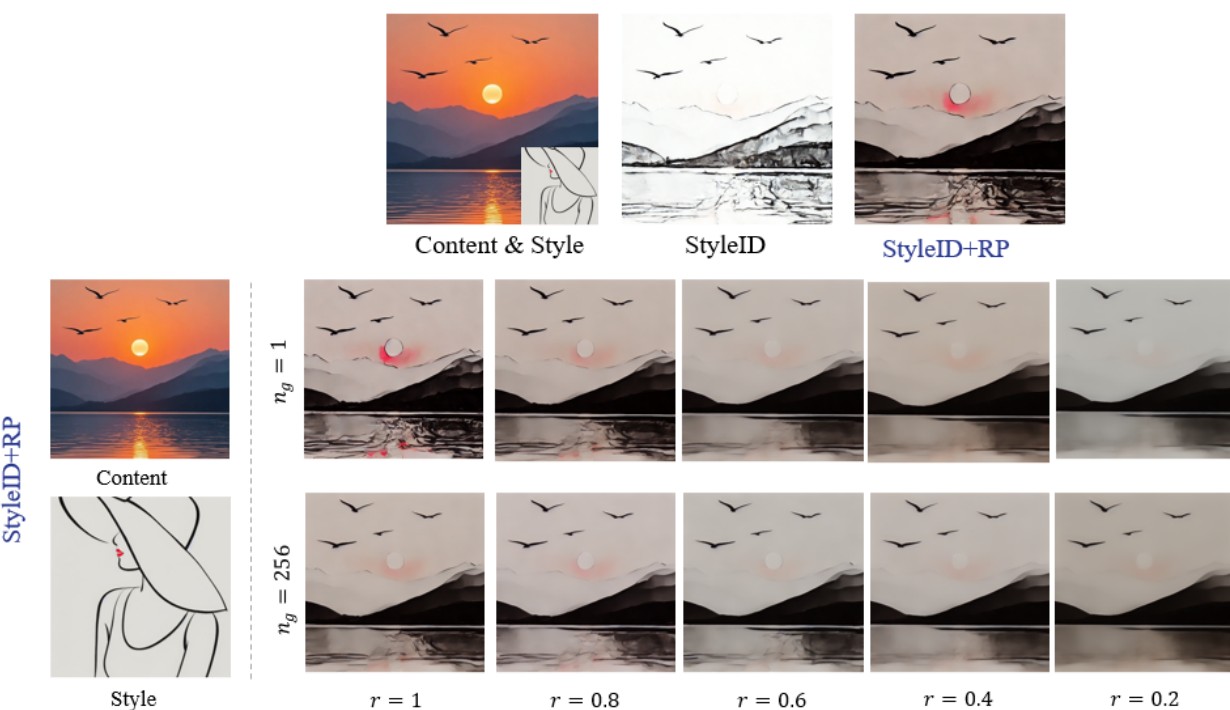

Figure 15: The stylized results derived by integrating our random projection module into StyleID, where the compression rate $r$ is decreased from 1 to 0.2 and the number $n_g$ of subgroups is increased from 1 to 256. Compared to StyleID, our method StyleID+RP outperforms notably in transferring weak style features (e.g., the lady's red lips), when the compression rate $r$ is sufficiently large.

## A.8 Comparison between our AdaAttN+RP and other well-recognized approaches

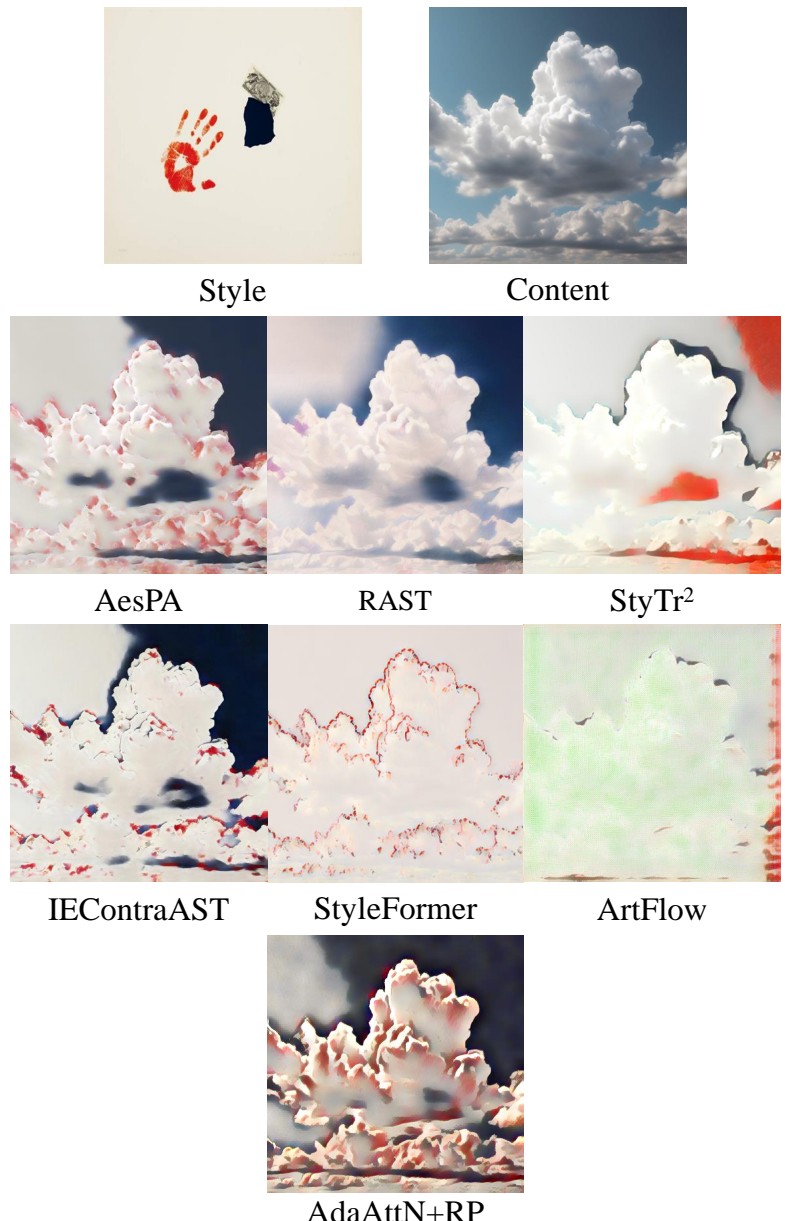

Figure 16: Stylized results derived by our AdaAttN+RP and other approaches. Among the approaches, our AdaAttN+RP stands out for its more saturated and evenly distributed red color.

Table 5: User Study Results for Figure 16

|  | AesPA | RAST | StyTr$^2$ | IEContraAST | StyleFormer | ArtFlow | AdaAttN+RP |
|---|---|---|---|---|---|---|---|
| Preference ratio ↑ | 0.13 | 0.00 | 0.00 | 0.07 | 0.02 | 0.00 | 0.78 |

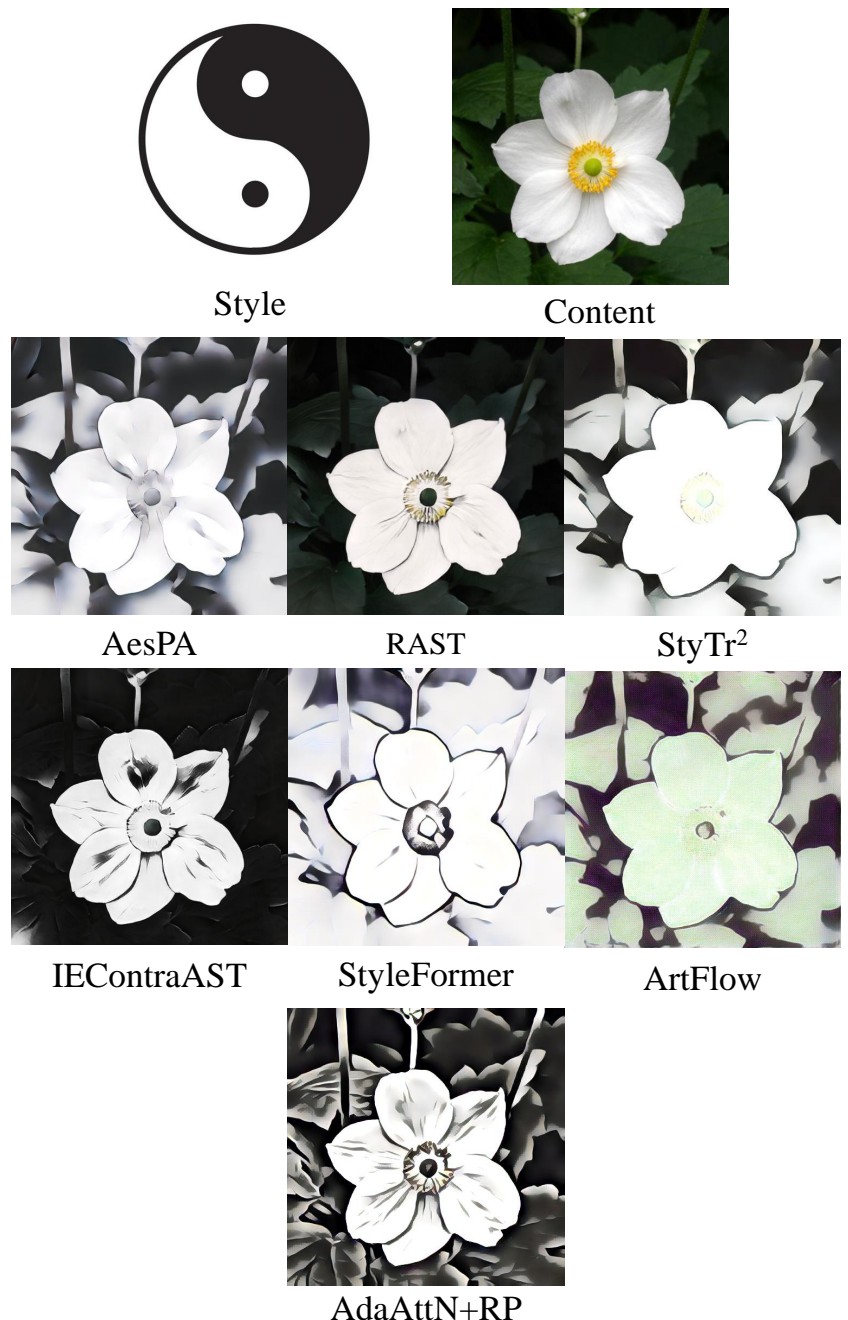

Figure 17: Stylized results derived by our AdaAttN+RP and other approaches. When rendering the textures of branches, leaves, and petals in a black-and-white style, our AdaAttN+RP outperforms all others.

Table 6: User Study Results for Figure 17

|  | AesPA | RAST | StyTr$^2$ | IEContraAST | StyleFormer | ArtFlow | AdaAttN+RP |
|---|---|---|---|---|---|---|---|
| Preference ratio ↑ | 0.08 | 0.13 | 0.00 | 0.01 | 0.05 | 0.00 | 0.73 |

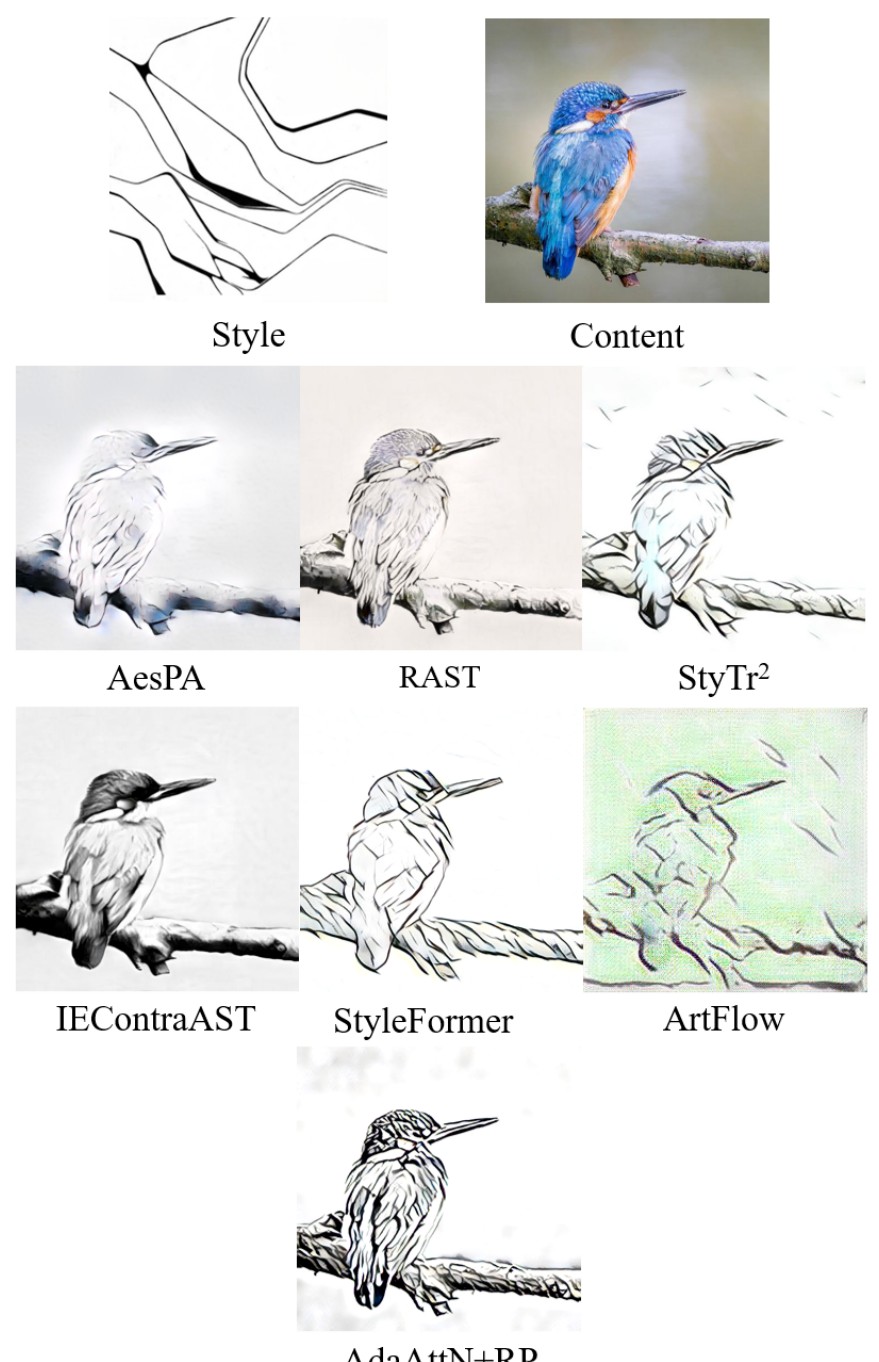

Figure 18: Stylized results derived by our AdaAttN+RP and other approaches. Our AdaAttN+RP achieves a good balance between style transfer and content preservation. In contrast, other approaches either overemphasize styles at the expense of content details, as seen with StyleFormer, or lack sufficient stylistic features, like RAST.

Table 7: User Study Results for Figure 18

|  | AesPA | RAST | StyTr$^2$ | IEContraAST | StyleFormer | ArtFlow | AdaAttN+RP |
|---|---|---|---|---|---|---|---|
| Preference ratio ↑ | 0.02 | 0.11 | 0.07 | 0.00 | 0.27 | 0.01 | 0.52 |

