# OpenReview forum: "Random Projection-Induced Gaussian Latent Features for Arbitrary Style Transfer"
_TMLR — Accepted by TMLR_

### Review · Reviewer_JV3C · 2025-10-26

**Summary Of Contributions:**

This paper addresses a well-known discrepancy in modern style transfer methods like AdaIN: the assumption that latent features follow a Gaussian distribution. The authors correctly point out that features from natural images are, in practice, often sparse, which violates this core assumption and can degrade transfer quality. To solve this, the paper proposes an approach based on random projections. Sparse, high-dimensional features are first mapped to a lower-dimensional space using random projection. Based on statistical theory, these projections become denser and more closely approximate a Gaussian distribution. Existing style transfer methods (like AdaIN or EFDM) are then applied in this new, more "Gaussian-friendly" latent space, where their underlying assumptions are better met. Finally, the stylized low-dimensional features are recovered back into the original high-dimensional feature space using techniques from compressed sensing. The paper provides experimental validation showing that this module improves performance for both AdaIN and EFDM, leading to results that better preserve content structure.

**Audience:**

Yes

**Audience Explanation:**

This paper should be of high interest to the TMLR audience, particularly researchers and practitioners in computer vision, generative models, and creative AI. The contribution is solid, well-motivated, and technically sound. The fact that the module is model-agnostic and avoids the need for retraining makes it highly practical and more likely to be adopted by the community. The paper clearly demonstrates its value by improving upon two different (and recent) state-of-the-art methods.

**Broader Impact Concerns:**

None.

**Claims And Evidence:**

Yes

**Claims Explanation:**

The authors provide experimental validation.

**Requested Changes:**

The authors state that the compressed sensing optimization is solved quickly in absolute terms, but this is vague. Could the authors please add a breakdown of the runtime for each component: the initial feature extraction (forward pass); the random projection; the stylization step; and the compressed sensing reconstruction step? This would clarify the practical overhead introduced by the proposed module compared to the baseline methods.

This new module introduces several new hyperparameters, such as the target dimension for the random projection and any parameters related to the compressed sensing solver. In general, how much tuning is required for good performance, and are there default recommended values for these hyperparameters?

Please double-check the manuscript's formatting against the TMLR journal style. The font used in the current draft does not appear to match the official TMLR template.

---

> ### Author Response · Authors · 2025-10-29
> **Response to Reviewer JV3C**
>
> Dear Reviewer JV3C,
>
> Thank you for taking the time to review our paper.  We have addressed all concerns raised by the three reviewers and revised the manuscript in line with their comments.  The revised sections mainly include the content marked in red, as well as Appendices A.4 and A.5. Below, we provide detailed responses to each of your questions.
>
> **Comment 1:** The authors state that the compressed sensing optimization is solved quickly in absolute terms, but this is vague. Could the authors please add a breakdown of the runtime for each component: the initial feature extraction (forward pass); the random projection; the stylization step; and the compressed sensing reconstruction step? This would clarify the practical overhead introduced by the proposed module compared to the baseline methods.
>
> **Response 1:**   As outlined in Section 3.3, we utilize the Fast Iterative Shrinkage-Thresholding Algorithm (FISTA) for compressed sensing reconstruction. This iterative algorithm exhibits linear complexity, specifically $\mathcal{O}(MN)$ per iteration, with $M\times N$ representing the dimensions of the sensing matrices.
>
> In the tables below, we present the time consumption for feature encoding, style transfer, and decoding within the conventional networks AdaIN (which shares the same framework as EFDM), AdaAttN, and StyleID. Additionally, we include the time required for the random projection and sparse reconstruction processes within our random projection (RP) module. It is evident that the time overhead introduced by our RP module is very little, particularly when compared to the substantial time savings achieved by eliminating the need for network retraining.  These results have been included in the revised manuscript (Appendix A.5).
>
> |AdaIN/EFDM|||\|Our RP module||
> |-|-|-|-|-|
> |Encoding|Style transfer|Decoding|\|Random projection|Sparse reconstruction|
> |136ms|18ms|22ms|196ms|203ms|
>
> |AdaAttN|||\|Our RP module||
> |-|-|-|-|-|
> |Encoding|Style transfer|Decoding|\|Random projection|Sparse reconstruction|
> |872ms|172ms|371ms|2401ms|2303ms|
>
>
> |StyleID|||\|Our RP module||
> |-|-|-|-|-|
> |Encoding|Style transfer|Decoding|\|Random projection|Sparse reconstruction|
> |10533ms|10ms|5603ms|590ms|626ms|
>
> **Comment 2:**  This new module introduces several new hyperparameters, such as the target dimension for the random projection and any parameters related to the compressed sensing solver. In general, how much tuning is required for good performance, and are there default recommended values for these hyperparameters?
>
> **Response 2:** For the sake of generality, we maintain the consistency of key parameters across all experiments.  Specifically, we set the compression ratio for random projection to  $r=M/N=0.8$. For  style transfer on projection features, we set the number of subgroups to $n_g=1$.  Regarding the sparse reconstruction algorithm FISTA, we assign a regularization parameter of $\lambda=0.5$ in Eq. (4) and limit the maximum number of iterations to 10.  As shown in Figure 4,  increasing the number of subgorups $n_g$ or decreasing the compression ratio $r$ yields a more pronounced stylistic effect,  at the expense of certain content loss. The two parameters can be flexibly  adjusted based on practical requirements.  We have incorporated these specifics   into our revised manuscript.

---

### Review · Reviewer_jf7m · 2025-11-11

**Summary Of Contributions:**

This paper proposes a simple plug-and-play module for arbitrary image style transfer based on random projection induced Guassian latents. The main idea is that deep encoder features are typically sparse and therefore violate the Gaussian assumptions underlying the AdaIN and EFDM formulations. This work first applies random projection to both content and style features to obtain lower-dimensional features that more closely resemble Gaussian distributions. Style transfer (AdaIN or EFDM) is then performed in this projected space. Finally, the stylized projection is mapped back to the original feature space via sparse recovery using compressed sensing. The method is training-free and can be integrated into AdaIN, EFDM, AdaAttN, and StyleID.

The paper claims improved performance, especially when style features are sparse, and provides qualitative comparisons, ablations on projection matrices and grouping, and quantitative evaluations with LPIPS, ArtFID, and user studies.

**Audience:**

Yes

**Audience Explanation:**

## Strength

- The motivation is clear and grounded in a real limitation of both AdaIN and EFDM: sparse deep features lead to poor statistics matching against Gaussian. The failure cases of the popular methods shown in the paper support this point well.
- The proposed module is lightweight and training-free, and integrates easily into many existing style transfer pipelines.
- The improvements on sparse-style cases (e.g., red-lip sketch, sparse point patterns) are convincing and consistent across methods.
- The ablations on compression ratio, projection matrix types, and grouping shows that the proposed method is effective in style transfer.

## Weakness

- The entire method is built on the claim that random projection makes sparse features “more Gaussian,” but there is no quantitative evidence for this. Some normality metrics would make the motivation much more solid.
- The method appears strongest on sparse style images. Improvements on dense or highly textured styles are less clear, and in some cases, only slightly different from the baseline. The work would benefit from a more explicit discussion of when the method is helpful and when it is not.

**Broader Impact Concerns:**

None.

**Claims And Evidence:**

Yes

**Claims Explanation:**

The core claim—that AdaIN and EFDM degrade when the style feature distribution is sparse—is well supported through the provided qualitative examples. The proposed module alleviates content loss in sparse-style settings, and the improvements show up across multiple approaches (AdaIN, EFDM, AdaAttN, StyleID). Quantitative metrics support the trend.

The paper does not provide quantitative evidence for the central premise that random projection makes features “more Gaussian.” Runtime overhead from sparse reconstruction is also not discussed.

**Requested Changes:**

- Quantify the “Gaussianization” assumption: this work claims that random projection makes sparse features approximately Gaussian, which requires more concrete evidence.
- Sparse reconstruction is non-trivial. The per-image latency and comparisons to AdaIN, EFDM, AdaAttN, and StyleID with and without the RP module is needed for concrete run-time analysis.

---

> ### Author Response · Authors · 2025-11-13
> **Response to Reviewer jf7m**
>
> Dear Reviewer jf7m,
>
> Thank you for taking the time to review our paper. We have addressed all concerns raised by the three reviewers and revised the manuscript in line with their comments. The revised sections mainly include the content marked in red, as well as Appendices A.4 and A.5. Below are our point-by-point answers to your questions.
>
>
> **Comment 1:** The entire method is built on the claim that random projection makes sparse features “more Gaussian,” but there is no quantitative evidence for this. Some normality metrics would make the motivation much more solid.
>
> **Response 1:**  Consider a random projection $y=Ax$, where $x\in R^n$, $y\in R^m$, $A\in R^{m\times n}$, $m\leq n$. We can derive the projection element $y_i$ that holds or approximates Gaussian distributions, based on  the following facts:
> 1) When  $A$  is a Gaussian matrix, with each entry $A_{i,j}$  i.i.d. drawn from $N(0,1)$,  the projection element $y_i$ is Gaussian according to *the closure property* of the *linear operations* of Gaussian variables.
> 2) When   $A$ is a binary or ternary matrix, namely $A_{i,j}\in\\{0,1\\}$ or $\\{0, \pm 1\\}$, by *the central limit theorem*, the projection element $y_i$  tends to be Gaussian when the original data dimension $n\rightarrow\infty$.
> 3)  When $A_{i,j}\in\\{0, \pm 1\\}$, by the the reference (Meckes, 2012), the projection vector $y$ will be close to independent Gaussian vectors, with the decreasing of the projection dimension $m$.
>
> **Comment 2:** The method appears strongest on sparse style images. Improvements on dense or highly textured styles are less clear, and in some cases, only slightly different from the baseline. The work would benefit from a more explicit discussion of when the method is helpful and when it is not.
>
> **Response 2:**  Note that what we focus on is the deep latent features (for style transfer) that have sparse distributions. In fact, most of the natural images have sparse-distributed latent features  due to the high continuity and correlation between image pixels.  This means that our method performs well not only with visually-*sparse* style images, but also with those images that are visually *dense*, such as the examples provided in Figures 8 and Figures 13.
>
> **Comment 3:** Sparse reconstruction is non-trivial. The per-image latency and comparisons to AdaIN, EFDM, AdaAttN, and StyleID with and without the RP module is needed for concrete run-time analysis.
>
> **Response 3:** Our random projection module takes very little time, at most about one or two seconds. For details, please see our Response 1 to Reviewer JV3C.

---

### Review · Reviewer_s4a3 · 2025-11-14

**Summary Of Contributions:**

The primary contribution of this paper is the introduction of a plug-and-play random projection module designed to address the sparsity issue inherent in deep convolutional features for style transfer. This module projects sparse style features into low-dimensional, Gaussian-like distributions, enhancing compatibility with existing methods and enabling the reconstruction of stylized features through compressed sensing, all without the need for network retraining.

Key advantages include its broad applicability across various style transfer frameworks, significant improvements in content preservation and style richness, as demonstrated by both qualitative and quantitative results, and efficient implementation with GPU acceleration.

However, potential limitations include the need for careful parameter tuning (e.g., random matrix type, compression rate) to achieve optimal performance, as well as a slight increase in computational cost due to sparse reconstruction, although this is minimal in practice.

**Additional Comments:**

This is a well-written and technically robust paper that addresses a significant gap in style transfer research. The plug-and-play design is a standout feature, as it lowers the adoption barrier across existing models. The integration with diffusion models and attention-based methods highlights the module’s versatility, which is especially valuable given the increasing popularity of these frameworks. The experimentation further validates their findings. By making the code publicly available, as promised, they will facilitate reproducibility and enable further research in this area.

**Audience:**

Yes

**Audience Explanation:**

The paper tackles the issue of sparse feature incompatibility in state-of-the-art methods, providing a generalizable solution that seamlessly integrates with popular frameworks. Researchers in generative models, computer vision, and representation learning will benefit from the insights on feature distribution alignment. Practitioners will appreciate the plug-and-play design, which enhances performance without the need for retraining, making it highly suitable for real-world applications.

**Broader Impact Concerns:**

No major ethical concerns require a Broader Impact Statement or additional addressing.

**Claims And Evidence:**

Yes

**Claims Explanation:**

1. Qualitative results visually demonstrate that RP-enhanced models outperform baselines in preserving details, transferring subtle style elements, and balancing content with style.

2. The RP models consistently achieve lower LPIPS/ArtFID scores, indicating better content fidelity and overall quality, and higher user preference ratios, reaching up to 87%.
3. The methodology is grounded in established theories, such as the Johnson-Lindenstrauss lemma for random projection and compressed sensing for reconstruction, ensuring technical rigor.

4. The authors also test multiple parameters, including matrix types and compression rates, to confirm the robustness of the approach.

**Requested Changes:**

(1) Please provide ablation studies on the impact of different sparse reconstruction algorithms. I am wondering if these different choices would affect the results slightly?

(2) While GPU acceleration is mentioned, including specific runtime comparisons would offer a clearer understanding of the practical feasibility of the approach.

(3) Another important aspect to consider is the performance of random projection in edge cases, such as when dealing with extremely dense features or very low compression rates. Addressing these scenarios would enhance the comprehensiveness of the work.

(4) Finally, offering guidance for parameter tuning could significantly improve usability. A concise table or flowchart outlining how to select the random matrix type, compression rate, and subgroup number based on input feature sparsity would be a valuable addition for users implementing the method.

---

> ### Author Response · Authors · 2025-11-18
> **Response to Reviewer s4a3**
>
> Dear Reviewer s4a3,
>
> Thank you for taking the time to review our paper. We have addressed all concerns raised by the three reviewers and revised the manuscript in line with their comments. The revised sections mainly include the content marked in red, as well as Appendices A.4 and A.5. Below, we provide point-by-point responses to your questions.
>
> **Comment 1:**  Please provide ablation studies on the impact of different sparse reconstruction algorithms. I am wondering if these different choices would affect the results slightly?
>
> **Response 1:** As suggested, besides the optimization-based algorithm FISTA, **we have further tested  the popular greedy-search-based algorithm OMP**(orthogonal matching pursuit) in Figure 14 (in the attachment A.4 of the revised manuscript).  It can be seen that the style-transfer difference between them is not obvious. This is because there is no significant difference in the reconstruction errors between the two algorithms.  In fact,  reconstruction errors are largely affected by the compression ratio $r$,  tending to increase with the decreased  $r$.  As shown in Figure 4, the increased reconstruction errors (random noise) tend to weaken content features and diversify stylized effect.
>
>  **Comment 2:**  While GPU acceleration is mentioned, including specific runtime comparisons would offer a clearer understanding of the practical feasibility of the approach.
>
> **Response 2:**  We would like to emphasize that the sparse reconstruction algorithm FISTA  not only has linear complexity ( $\mathcal{O}(MN)$ per iteration)  but also allows fast implementation on GPU. **The runtime comparison has been provided in our Response 1 to Reviewer JV3C**. It is seen that  the algorithm runs very fast,  taking at most  one or two seconds for the sparse reconstruction of a feature map.
>
>  **Comment 3:**  Another important aspect to consider is the performance of random projection in edge cases, such as when dealing with extremely dense features or very low compression rates. Addressing these scenarios would enhance the comprehensiveness of the work.
>
>  **Response 3:** **1) Regarding extremely dense features:**  Note that what we focus on is the deep latent features (for style transfer) that have sparse distributions. In fact, most of the natural images have sparse-distributed latent features  due to the high continuity and correlation between image pixels.  This means that our method performs well not only with visually-*sparse* style images, but also with those images that are visually *dense*, such as the examples provided in Figures 8 and Figures 13.   **2) Regarding low compression ratios:**  In Figure 4, we have decreased the compression ratio from $r=1$ to $0.2$. With the decreasing of the compression ratio $r$,  reconstruction errors (random noise) tend to become large, then  weakening content features and diversifying stylized effects.
>
>  **Comment 4:** Finally, offering guidance for parameter tuning could significantly improve usability. A concise table or flowchart outlining how to select the random matrix type, compression rate, and subgroup number based on input feature sparsity would be a valuable addition for users implementing the method.
>
>  **Response 4:**   For the sake of generality, we maintain the consistency of key parameters across all experiments.  Specifically, we set the compression ratio for random projection to $r=M/N=0.8$. For style transfer on projection features, we set the number of subgroups to $n_g=1$.  Regarding the sparse reconstruction algorithm FISTA, we assign a regularization parameter of $\lambda=0.5$ in Eq. (4) and limit the maximum number of iterations to 10.  As shown in Figure 4,  increasing the number of subgroups $n_g$ or decreasing the compression ratio $r$  tends to yield a more pronounced stylistic effect,  at the expense of certain content loss. The two parameters can be flexibly adjusted based on practical requirements.  We have incorporated these specifics  into our revised manuscript.

---

> ### Author Response · Authors · 2025-12-27
> **Recommendation for the selection of key parameters  (Response to the previous Comment 4)**
>
> Dear **Reviewer s4a3:**
>
> Thank you for your further comments. We regret that our previous response did not fully meet your expectations. We have carefully revisited your review comments, **confirmed that the experiments required in Comments 1–3 have been carried out, and identified that the issue may lie in your Comment 4**. You suggested adding a parameter tuning table or flowchart for ease of parameter adjustment and practical use, whereas we only provided  fragmented textual descriptions previously. To address this, we have further summarized the key parameter tuning details in the appendix (A.6) of the revised manuscript.
>
> By the way, we have provided the code at the link: https://anonymous.4open.science/r/Random-Projection-Induced-Gaussian-Latent-Features-for-Arbitrary-Style-Transfer-7844
>
> **Recommendation for the selection of key parameters:**
>
> 1) **Random projection matrices:**  In most cases, Gaussian matrices with zero mean and unit variance deliver satisfactory performance. If Gaussian matrices fail to meet requirements (e.g., for dense content features as illustrated in Figure 3),   binary \{0, 1\} matrices  may yield better performance.
>
> 2) **Compression rate $r=M/N$:**  According to compressed sensing theory, perfect sparse recovery requires the compression rate to satisfy $M>2k$, where $k$ denotes the feature sparsity, namely the number of nonzero entries in the $N$-dimensional feature vectors.  For style transfer tasks, besides recovery accuracy, we also need to consider the perceptual quality of the stylized results. Empirically, as illustrated in Figure 4,  setting $r\geq 0.6$ typically yields desirable performance.  A lower compression ratio $r$ tends to introduce more recovery errors (noise) , thereby degrading the content feature quality.
>
> 3) **Number $n_g$ of subgroups in projected features:**  For simplicity,  $n_g$ can be set to 1 in most cases.   As observed in Figure 4, increasing $n_g$ from 1 to 256 leads to slight variations in stylized effects, and such variations are far less pronounced than those caused by the compression rate $r$.
>
> 4) **Sparse recovery algorithms:** We recommend two commonly-used algorithms: FISTA and OMP. For FISTA, in our experiments the regulation parameter is set to $\lambda=0.5$ and the iteration number is limited to 10; and for OMP, the sparsity parameter $k$ can be determined based on feature sparsity.
>
> Thank you for your consideration. Should you have any remaining questions or need further clarification, please let us know.
>
> Sincerely,
>
> The authors

---

> ### Author Response · Authors · 2025-12-29
> **Response to the further comments of Reviewer s4a3**
>
> Dear **Reviewer s4a3**,
>
> Thank you very much for your insightful comments, which are helpful for improving our work. In response to your comments, we have further revised the manuscript (i.e., discussing the potential limitation in the end of the experiments section), and supplemented additional experiments in Appendices A.7 and A.8 (**Figures 15-18**). We address and respond to each of your comments in detail below.
>
> **Comment 1:** I highly appreciate the authors’ further responses. In fact, a holistic assessment of this work reveals that its theoretical innovation is rather limited. While it does offer a valuable practical perspective for researchers, the experimental section is notably deficient in comparisons against state-of-the-art methods. Therefore, I encourage the authors to present more detailed parameter descriptions in the experimental section, so that the contributions of this work can be highlighted.
>
>
> **Response 1:** Thank you for your recognition of our contributions to practical applications. As you noted, instead of theoretical innovation, our research aims to propose a simple yet effective method to improve the performance of existing style transfer approaches. **The experiments illustrated in Figures 1-17 have demonstrated that the proposed method achieves notable performance improvements when integrated into the state-of-the-art approaches**, including AdaIN, AdaAttN, EFDM, and the diffusion model-based StyleID, particularly when images exhibit sparse feature distributions.
>
> **As you suggested**,  **we have further investigated the performance of the three style transfer approaches (AdaAttN, EFDM, and StyleID) in Figures 15-17, Appendix A.7**, by varying the two key parameters: the compression rate $r=M/N$ and the number $n_g$ of subgroups. The results demonstrate that as the compression ratio $r $ decreases, the content quality tends to degrade while the stylized effect tends to be enhanced, owing to the introduction of more recovery errors (i.e. noise). Similarly, increasing the subgroup number $n_g$ from 1 to 256 slightly boosts stylization effects. **Regarding parameter tuning, simply setting $r=1$ and $n_g=1$ suffices for our method to achieve notable performance gains over existing approaches.**
>
>
> **Comment 2:** The abstract claims that “...projecting the sparse features into lower dimensions via random projection”. Here, a key question arises: do the “sparse features” refer to the features generated by the encoder (fc or fcp)? And are these features truly sparse? As we know, encoder-derived features are typically dense but possess sparse potential that can be exploited through appropriate transformations.
>
> **Response 2:** This question is of great importance.  **As suggested, we have discussed this issue in the Appendix A.8.** The discussion is detailed as follows.
>
> **These encoder features indeed tend to exhibit sparse distributions**, for two main reasons. First, most natural images contain relatively large smooth regions, which facilitate the generation of sparse feature distributions. Second, notably, these encoders do not reduce the input image dimensions to a considerably lower level. Specifically, unlike conventional encoders designed for dimensionality reduction, the encoders in AdaIN, AdaAttN and EFDM *increase* the feature dimension from the channel size of [3, 512, 512] to [512, 64, 64] (>2,000,000), whereas StyleID reduces the feature dimension from the channel size of [3, 512, 512] to [4, 64, 64] (>10,000). These feature dimensions are substantially larger than the conventional encoder features generated in typical VAE and GAN, which usually have dimensions on the order of *hundreds*. Such high dimensionality inevitably leads to feature redundancy, thereby resulting in sparse structures.
>
>
> To quantify the sparsity of these encoder features, as illustrated in **Figure 18**, **we have further investigated their *kurtosis*, a canonical statistical metric for quantifying the sparsity of random variables**. The kurtosis is defined as $\mathbb{E}(\frac{x_i-\mu}{\sigma})^4$, where $x_i$ denotes the $i$-th elements of a vector $x$, and $\mu$ and $\sigma$ are their mean and standard deviation. A larger kurtosis value indicates a sparser distribution. If the kurtosis value of a variable is larger than that (equal to 3) of Gaussian variables, the variable can be roughly regarded as following a sparse (or called heavy-tailed) distribution.  From Figure 18, it is seen that the kurtosis of AdaIN features is  higher than that of StyleID features, which can be attributed to their higher feature dimensions. **For most images, the kurtosis values of both feature types are notably higher than the Gaussian baseline (i.e., 3), verifying that they exhibit sparse distributions.**

---

> > ### Comment · Reviewer_s4a3 · 2025-12-30
> > **Thank you for the detailed response**
> >
> > Thank you to the authors for the detailed response. These observations provide valuable references for the application field, and I have no further questions.

---

> ### Author Response · Authors · 2025-12-29
>
> **Comment 3:** This work demonstrates the experimental results using matrix (\mathcal{A}_C). It is reasonable to anticipate that the performance using matrix (\mathcal{A}_S) would be relatively inferior. Nevertheless, readers (and me) are more concerned about the outputs generated by different combinations of parameters and optional operations involved in this work, as such comprehensive results would enable a more holistic assessment of the work’s value.
>
> **Response 3:** As suggested, we have provided additional experiments in  Appendix A.7 (Figures 15-17), as discussed   in our previous Response 1.
>
>
> **Comment 4:** Besides, we are also concerned the limitations of this work. How does this work perform on extremely sparse images? What are the predictable reasons for the potential degradation of this method? Please provide more insightful results and discussions.
>
> **Response 4:** The style images we tested include extremely sparse examples, such as the curves, spots, and character sketches illustrated in Figures 5–7. Excelling at handling extremely sparse features is one of our major advantages.
> **As suggested, we have discussed the potential limitation of our method in the final part of the experiments section, which is restated as follows.**
>
> For our random projection module, the main advantage is that it can prevent the loss of critical features in either content or style, when the latent features exhibit extremely sparse distributions. Typical examples include the body contour (content features) of the bird in Figure 1 and the red lip (style features) of the lady in Figure 7.
> **In practical scenarios, however, the feature loss of content or style caused by existing style transfer approaches may be imperceptible to human vision, resulting in an unobvious performance gain of our method.** Nevertheless, even without improved perceptual quality, **our module will not degrade the quality**, since compressed sensing enables reliable feature reconstruction from random projections when the compression ratio is not extremely low. This indicates that our  module can be universally integrated into existing style transfer approaches without any concern about side effects.

---

### Decision · Action_Editor_azWC · 2026-01-07

**Recommendation:** Accept as is

**Audience:**

Yes

**Audience Explanation:**

Style transfer is a longstanding area of interest and application of generative models. This paper addresses a known issue and as such will be interesting for that community. It also serves as a simple yet effective baseline for future works.

**Claims And Evidence:**

Yes

**Claims Explanation:**

This paper proposes a new approach for style transfer. The approach is simple to use and does not require training, relaxing gaussianity assumptions on the features made by prior approaches. All the reviewer leaned towards acceptance, agreeing that the paper is clear and sufficiently novel, although more extensive baselines comparison was called out as a potential improvement point. Nevertheless, due to the simplicity of the approach and the experiments demonstrating its positive effect, the paper is recommended for acceptance.